# THOR: Tool-Integrated Hierarchical Optimization via RL for Mathematical Reasoning

**Qikai Chang[1], Zhenrong Zhang[1,2], Pengfei Hu[1], Jun Du[1]\*, Jiefeng Ma[1,2], Yicheng Pan[1]**
**Jianshu Zhang[2], Quan Liu[2], Jianqing Gao[2]**

[1]University of Science and Technology of China, [2]iFLYTEK Research

## ABSTRACT

Large Language Models (LLMs) have made remarkable progress in mathematical reasoning, but still continue to struggle with high-precision tasks like numerical computation and formal symbolic manipulation. Integrating external tools has emerged as a promising approach to bridge this gap. Despite recent advances, existing methods struggle with three key challenges: constructing tool-integrated reasoning data, performing fine-grained optimization, and enhancing inference. To overcome these limitations, we propose THOR (**T**ool-Integrated **H**ierarchical **O**ptimization via **R**L). First, we introduce TIRGen, a multi-agent based pipeline for constructing high-quality datasets of tool-integrated reasoning paths, aligning with the policy and generalizing well across diverse models. Second, to perform fine-grained hierarchical optimization, we introduce an RL strategy that jointly optimizes for both episode-level problem solving and step-level code generation. This is motivated by our key insight that *the success of an intermediate tool call is a strong predictor of the final answer's correctness*. Finally, THOR incorporates a self-correction mechanism that leverages immediate tool feedback to dynamically revise erroneous reasoning paths during inference. Our approach demonstrates strong generalization across diverse models, performing effectively in both reasoning and non-reasoning models. It further achieves state-of-the-art performance for models of a similar scale on multiple mathematical benchmarks, while also delivering consistent improvements on code benchmarks. Our code will be publicly available at https://github.com/JingMog/THOR.

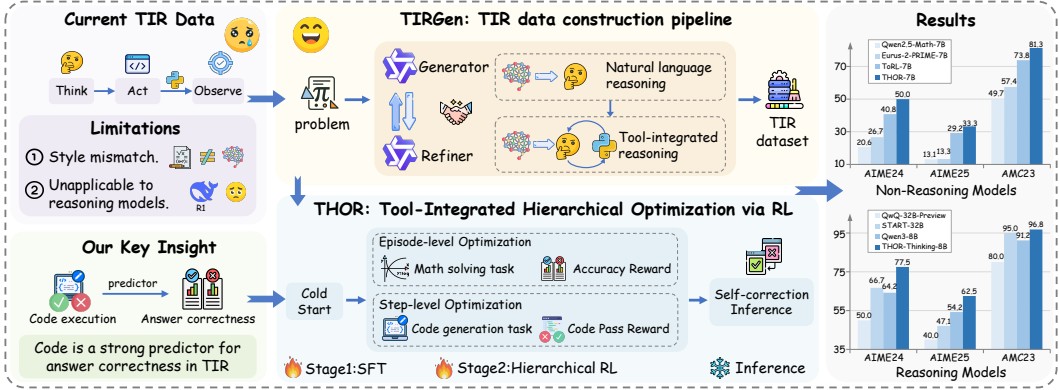

Figure 1: An overview of our method. The left panel depicts the motivation and challenges, the middle highlights our proposed solution with the TIRGen data construction pipeline and the THOR hierarchical RL framework, and the right panel reports experimental results.

---

\*Corresponding author: Jun Du (jundu@ustc.edu.cn).

# 1 INTRODUCTION

Large Language Models (LLMs) have achieved remarkable progress, increasingly exhibiting human-like capabilities such as thinking, reflection, and self-correction. They have shown significant improvements in mathematical reasoning, code generation, and autonomous agent tasking (Jaech et al., 2024; Guo et al., 2025; Yang et al., 2025; Team et al., 2025).

Recent tool-free methods for enhancing LLMs' mathematical reasoning can be broadly categorized into search-based methods (Besta et al., 2024; Zhang et al., 2025; Hu et al., 2025) and reinforcement learning (RL) paradigms (Yu et al., 2025; Yue et al., 2025b; Luo et al., 2026). Despite notable progress, these approaches remain constrained by a fundamental weakness of LLMs. As probabilistic next-token predictors, they inherently struggle with high-precision tasks (Chen et al., 2022), such as numerical computation, equation solving, symbolic manipulation (Pan et al., 2025), and formal proofs (Lewkowycz et al., 2022). In contrast, programmatic reasoning excels in these domains. Therefore, integrating the semantic reasoning capabilities of LLMs with the precise and verifiable execution of external code-based tools provides a promising pathway to overcome these limitations.

Tool-Integrated Reasoning (TIR) has emerged as a powerful paradigm for enhancing LLM reasoning by enabling them to leverage external tools to augment reasoning (Gou et al., 2023; Li et al., 2025a). Despite considerable efforts, three core challenges remain: **constructing TIR data**, **performing fine-grained optimization**, and **enhancing inference**. (1) For **constructing TIR data**, a common approach is to synthesize tool-use data by prompting external powerful models (e.g., GPT-4o) (Gou et al., 2023). However, for reasoning models such as DeepSeek-R1, prompting alone often fails to elicit effective tool use (Guo et al., 2025; Li et al., 2025a). While techniques like START (Li et al., 2025a) explicitly inject code prompts into the thinking process, purely rule-based approaches struggle to identify suitable insertion positions. Therefore, existing TIR data construction methods suffer from style mismatches and poor applicability to reasoning models. (2) For **performing fine-grained optimization**, current research primarily employs either SFT or RL. SFT-based methods, like Toolformer (Schick et al., 2023) and Aimo-2 (Moshkov et al., 2025), require large-scale, high-quality demonstration data and often suffer from poor generalization. Existing RL methods (Mai et al., 2025; Li et al., 2025b; Feng et al., 2025) typically optimize at the episode-level, overlooking fine-grained updates on specific steps. Although RL is a more scalable alternative, it faces severe sparse reward problems, particularly in long reasoning chains. (3) For **enhancing inference**, existing methods typically interleave tool calls directly with natural language reasoning in a single pass, thereby overlooking the role of immediate tool feedback in reasoning.

To address these challenges, we propose THOR, a tool-integrated framework designed to enhance the reasoning ability of LLMs. (1) For **constructing TIR data**, in order to efficiently generate policy-aligned TIR data, we propose TIRGen, an generator-refiner-based data construction pipeline. The generator is responsible for generating natural language reasoning steps, while the refiner evaluates whether steps can be transformed into executable code and interacts with an external executor to refine the reasoning. This iterative process yields a TIR dataset that is naturally aligned with the generator's policy and broadly applicable across diverse models and tools. (2) For **performing fine-grained optimization**, we are motivated by the key insight that *the success of an intermediate tool call is a strong predictor of the final answer's correctness*. Our experiments later confirm this insight. Based on this, we introduce a hierarchical RL strategy that combines episode-level and step-level optimization. At the episode-level, we directly optimize for the correctness of the final answer. Concurrently, at the step-level, we apply fine-grained optimization to execution failure steps, specifically enhancing the model's code generation ability. (3) For **enhancing inference**, we propose a self-correction mechanism that leverages immediate feedback from tools to dynamically revise its CoT during inference. When code invocation fails, it backtracks and explores alternative reasoning paths, thereby significantly enhancing the model's reasoning robustness and overall performance.

We evaluate our method on diverse challenging and widely-used benchmarks, including MATH500 (Hendrycks et al., 2021), AIME 2024 & 2025, AMC, Minerva Math (Lewkowycz et al., 2022), and Olympiad Bench (He et al., 2024). THOR establishes a new state-of-the-art (SOTA) result among models of comparable size across architectures and scales, while reducing inference overhead. It further improves performance on code generation benchmarks HumanEval, MBPP (Liu et al., 2023), and LiveCodeBench, validating the effectiveness and generalizability of our approach.

Our primary contributions are as follows: 1) Tool-Integrated Data Construction Pipeline. We introduce TIRGen, a pipeline for generating TIR data, applicable across diverse models, and better aligned with the preferences of the policy model. 2) Hierarchical Optimization. We propose a hierarchical reinforcement learning approach that combines episode-level and step-level optimization. 3) Self-correction Inference Enhancement. We introduce a self-correction mechanism that leverages immediate tool feedback to revise reasoning steps during inference. 4) Superior Performance and Broad Generalization. Our approach generalizes across reasoning and non-reasoning models, achieving competitive results on mathematical benchmarks and consistent gains on code tasks.

## 2 METHODOLOGY

### 2.1 PROBLEM FORMULATION

In the context of tool-integrated reasoning, an LLM solves mathematical problems by interleaving natural language reasoning with tool invocations. Specifically, we formulate an LLM, parameterized by $\theta$, as a policy $\pi_\theta$. Given a problem $q$ and a corresponding instruction $I$, this policy $\pi_\theta$ autoregressively generates an entire interaction trajectory $\tau$, which is an alternating sequence of thoughts, actions, and observations:

$$\tau = (r^1, a^1, o^1, ..., r^t, a^t, o^t, ..., r^{n-1}, a^{n-1}, o^{n-1}, r^n), \quad (1)$$

where $r^t$ is a step of natural language reasoning, $a^t$ is an action of tool call, $o^t$ is the observation returned by the external execution environment after executing action $a^t$, and $n$ is the number of reasoning steps. This process is formulated as an iterative think-act-observe loop. The model incorporates the new observation $o^t$ into its context to inform the generation of the subsequent thought $r^{t+1}$ and action $a^{t+1}$. This cycle continues until the model produces the final answer within its last thought $r^n$, thereby concluding the trajectory. Formally, the likelihood of generating a specific trajectory $\tau$ is factorized as:

$$P_{\pi_\theta}(\tau \mid q, I) = P_{\pi_\theta}(r^n \mid q, I, \mathcal{H}^{1:n-1}) \prod_{t=1}^{n-1} \underbrace{P_{\pi_\theta}(r^t \mid q, I, \mathcal{H}^{1:t-1})}_{\text{Thought}} \underbrace{P_{\pi_\theta}(a^t \mid r^t, q, I, \mathcal{H}^{1:t-1})}_{\text{Action}}, \quad (2)$$

where $\mathcal{H}^{1:t-1} = \{r^1, a^1, o^1, \ldots, r^{t-1}, a^{t-1}, o^{t-1}\}$ denotes the history of the previous interactions. Each term is modeled as a product of token-level probabilities.

### 2.2 TIRGEN: TIR DATA GENERATION PIPELINE

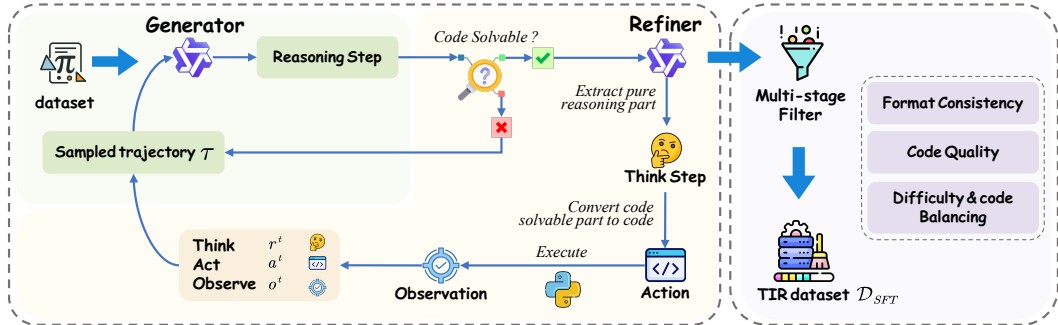

Figure 2: The TIR data construction pipeline. In this pipeline, the Generator agent generates reasoning steps. The Refiner agent identifies tool-executable steps and converts them into tool-augmented reasoning steps. After multi-stage filtering, we obtain the cold start dataset $\mathcal{D}_{SFT}$.

Existing methods for TIR highlight a significant need for high-quality training data. Most approaches rely on simply prompting external large models to synthesize TIR data for non-reasoning models (Gou et al., 2023; Li et al., 2025b), but these approaches fail to generalize effectively to reasoning models such as DeepSeek-R1. Although START (Li et al., 2025a) constructs long-CoT TIR data using a rule-based prompt-hint approach, the resulting trajectories often contain redundant

---

**Algorithm 1** TIRGen: TIR Data Generation Pipeline

---

1: **Input:** Generator model $\pi_{\text{gen}}$, Refiner model $\pi_{\text{refiner}}$, Dataset $\mathcal{D}_q$, Code interpreter sandbox $S$.
2: **Initialize:** Raw cold start dataset $\mathcal{D}_{raw} \leftarrow \emptyset$
3: **for** question $q \in \mathcal{D}_q$ **do**
4:     Initialize trajectory $\tau \leftarrow (q)$
5:     **while** not IsSolved$(\tau)$ **do**
6:         $r^t \sim \pi_{\text{gen}}(\cdot \mid \tau)$, $|r^t| \leq L_{\text{step}}$         ▷ Generator $\pi_{\text{gen}}$ generates a reasoning step
7:         **if** JudgeCodeSolvable$(r^t)$ **then**     ▷ Identify operation solvable with code by $\pi_{\text{refiner}}$
8:             $r^t_{\text{logic}} \leftarrow \text{ExtractLogic}_{\pi_{\text{refiner}}}(r^t)$     ▷ Step 1: Extract the pure reasoning part
9:             $a^t \leftarrow \text{ConvertToCode}_{\pi_{\text{refiner}}}(r^t, r^t_{\text{logic}})$    ▷ Step 2: Convert calculation part to code
10:           $o^t \leftarrow S(a^t)$                ▷ Step 3: Execute code to get observation
11:           $\tau \leftarrow \tau \oplus (r^t_{\text{logic}}, a^t, o^t)$
12:         **else**
13:           $\tau \leftarrow \tau \oplus (r^t)$
14:     $\mathcal{D}_{raw} \leftarrow \mathcal{D}_{raw} \cup \{\tau\}$
15: $\mathcal{D}_{SFT} \leftarrow \text{MultiStageFilter}(\mathcal{D}_{raw})$       ▷ Filter for format, code quality and difficulty
16: **Return** $\mathcal{D}_{SFT}$

---

code invocations. Therefore, existing TIR data construction methods face critical shortcomings, including style mismatches between the generated data and policy models, as well as poor scalability to reasoning models. To overcome these limitations, we introduce TIRGen, an automated TIR data synthesis pipeline based on an generator-refiner framework, as illustrated in Figure 2.

In this framework, the Generator generates a natural language reasoning step with a maximum length of $L_{\text{step}}$. The Refiner agent then evaluates this step to identify operations that can be more efficiently solved with code, such as numerical calculations or equation solving. Upon identifying such an operation, the Refiner converts it into an executable Python code snippet while preserving the Generator's original reasoning logic. The Refiner then interacts with a code interpreter to obtain a precise execution result, which replaces the original operation and produces a code-augmented reasoning step. The Generator resumes reasoning from this revised step, and the iterative cycle continues until a complete solution is reached, as described in Algorithm 1. This design yields two key advantages:

- **Policy Alignment.** Because the Refiner observes only isolated reasoning steps without the full problem statement and final answer, the synthesized data directly reflects the Generator's intrinsic reasoning preferences. This ensures that the data remains in-distribution, thereby mitigating the performance degradation commonly caused by training on out-of-distribution data (Gudibande et al., 2023; Chen et al., 2024).
- **Reduced Reliance on Large-scale Models.** The Generator is responsible for high-level mathematical reasoning, while the Refiner requires only basic instruction-following and code-generation skills. This division of labor decomposes the complex task into subproblems that are collaboratively solved by the two agents.

After sampling, we apply a multi-stage filtering procedure: (1) Format Consistency. We remove samples with erroneous tool calls or incorrect formats, and enforce that final answers are wrapped in \boxed{}. (2) Code Quality. We discard code that fails to execute or contains only trivial operations. Retained examples must include library invocations (e.g., sympy, numpy) or control flows (loops or branches). (3) Difficulty & Call-Round Balancing. To ensure diversity in problem complexity and tool invocation rounds, we stratify samples by the number of code calls and apply moderate down-sampling within each subset. Additionally, we remove instances solvable by a pure CoT baseline, ensuring that all retained examples require tool integration. This comprehensive procedure yields the final dataset for the cold start phase, denoted as $\mathcal{D}_{SFT}$.

## 2.3 HIERARCHICAL RL TRAINING STRATEGY

### 2.3.1 COLD START

We initialize our model with a cold start procedure (Guo et al., 2025), utilizing the dataset $\mathcal{D}_{SFT}$ generated by TIRGen. The primary objective of this stage is to equip the model with the fundamental

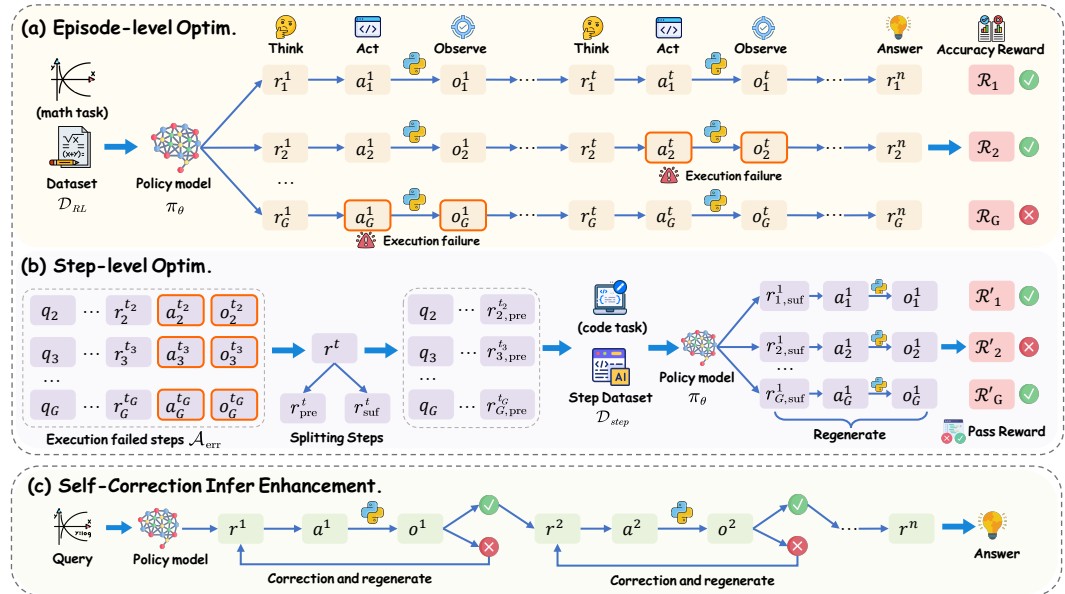

Figure 3: A hierarchical optimization framework comprising (a) episode-level RL for mathematical problem solving and (b) step-level optimization for code generation. In addition, we introduce (c) a self-correction mechanism for online error correction during inference.

patterns of tool invocation. This initial training is crucial, as reasoning models often struggle to invoke code effectively before a cold start. To this end, we perform supervised fine-tuning on the base model $\tilde{\pi}_\theta$ with the loss function as follows:

$$\mathcal{L}_{\text{SFT}}(\theta) = \mathbb{E}_{(q,I,y)\sim\mathcal{D}_{SFT}}\Big[ -\sum_{t=1}^{T} \log \tilde{\pi}_\theta(y_t \mid q, I, y_{0:t-1})\Big], \tag{3}$$

where $T$ is the length of response. In this way, we obtained the policy model $\pi_\theta$ after the cold start.

### 2.3.2 HIERARCHICAL REINFORCEMENT LEARNING

In the hierarchical optimization phase, we employ RL to refine the model's policy for invoking tools to solve complex problems. Conventional RL methods perform optimization at the episode-level, relying on a sparse reward signal derived solely from the correctness of the final outcome (Mai et al., 2025; Li et al., 2025b; Feng et al., 2025; Tao et al., 2025). Our empirical analysis reveals a crucial insight: *the success of an intermediate tool call is a strong predictor of the final answer's correctness*. Motivated by this finding, we propose a hierarchical reinforcement learning framework that combines coarse-grained, episode-level optimization to enhance the model's problem-solving ability with fine-grained, step-level optimization to improve code generation for individual tool calls, as shown in Figure 3. The observation is validated in Appendix C.1.

**Episode-level Optimization.** At the episode-level optimization, we adopt GRPO (Shao et al., 2024) and use the correctness of the final answer as the reward to enhance the tool invocation strategy of the model, as shown in Figure 3(a). The complete episode-level objective is:

$$\mathcal{L}_{\pi_\theta}^{\text{epis}}(\theta) = \mathbb{E}[q \sim \mathcal{D}_{RL}, \{s_i\}_{i=1}^{G} \sim \pi_\theta(S|q)]\frac{1}{G}\sum_{i=1}^{G}\Big(\frac{1}{\sum_{t=1}^{|s_i|} I(s_{i,t})}\sum_{t:I(s_{i,t})=1}^{|s_i|}$$
$$\min\Big(\frac{\pi_\theta(s_i|q)}{\pi_{\theta_{\text{old}}}(s_i|q)}A_i, \text{clip}(\frac{\pi_\theta(s_i|q)}{\pi_{\theta_{\text{old}}}(s_i|q)}, 1-\varepsilon_{\text{low}}, 1+\varepsilon_{\text{high}})A_i\Big)\Big) + \alpha\mathcal{L}_{\text{NLL}}(\theta), \tag{4}$$

where $G$ denotes the group size, and $A_i$ is the advantage of response $s_i$, defined as $A_i = \frac{\mathcal{R}_i - \text{mean}(\mathcal{R})}{\text{std}(\mathcal{R})}$. The reward is given by $\mathcal{R}_i = 1$ if answer is correct, and $0$ otherwise. The indicator function

$I(s_{i,t}) = 1$ indicates that the token $s_{i,t}$ is generated by $\pi_\theta$ rather than observed from an external executor. Additionally, to better exploit successful trajectories, we incorporate a language modeling loss $\mathcal{L}_{\text{NLL}}$, weighted by a coefficient $\alpha$, applied to examples with positive advantage $\mathcal{T}_{\text{pos}}$, directly reinforcing the likelihood of correct samples during RL training (Yue et al., 2025b).

$$\mathcal{L}_{\text{NLL}}(\theta) = -\frac{1}{\sum_{s_i \in \mathcal{T}_{\text{pos}}} |s_i|} \sum_{s_i \in \mathcal{T}_{\text{pos}}} \log \pi_\theta(s_i|q). \tag{5}$$

To stabilize training, we filter out trajectories with failed code executions during episode-level optimization to avoid inappropriate penalties. Since execution failures may arise not only from model-generated errors but also from environment issues or sandbox limitations.

**Step-level Optimization.** After sampling full trajectories, we perform step-level optimization to correct code errors using execution feedback, as illustrated in Figure 3(b). This stage focuses specifically on reasoning steps that resulted in failed actions $\mathcal{A}_{\text{err}}$. First, we construct a step-level optimization dataset $\mathcal{D}_{step}$. For each failed step, we treat it as a query and generate group rollouts, where the execution correctness of the generated code provides the reward signal. Consequently, it is crucial to ensure that the rollouts within each group cover diverse execution outcomes. Next, we describe the construction of $\mathcal{D}_{step}$ and its corresponding optimization method.

Within a think-act-observe tuple $(r^t, a^t, o^t)$, a failed execution inherently indicates an error exists in the action $a^t$. Moreover, since $a^t$ is conditioned on $r^t$, fixing $r^t$ limits the diversity of generated actions. To overcome this problem, we implement a backtracking procedure. We partition the erroneous thought $r^t$ into a prefix $r^t_{\text{pre}}$ and a suffix $r^t_{\text{suf}}$ of length $L_{\text{suf}}$. Subsequently, we condition the model on the history up to $r^t_{\text{pre}}$ and regenerate a new reasoning suffix and action. This procedure produces the dataset $\mathcal{D}_{step}$ for fine-grained step-level optimization:

$$\mathcal{D}_{step} = \{\text{pref}(\tau, t) \mid a^t \in \mathcal{A}_{\text{err}}, \tau \in \mathcal{T}\}, \tag{6}$$

$$\text{pref}(\tau, t) = (q, r^1, a^1, o^1, ..., r^{t-1}, a^{t-1}, o^{t-1}, r^t_{\text{pre}}), r^t = r^t_{\text{pre}} \oplus r^t_{\text{suf}}, \tag{7}$$

This formulation defines a fine-grained code generation task: given a reasoning prefix $\text{pref}(\tau, t)$, the model must correctly regenerate the subsequent reasoning suffix $\hat{r}^t_{\text{suf}}$ and action $\hat{a}^t$. At this stage, each sample contains a single think-act-observe loop, and the reward is computed from execution correctness. We optimize the policy with the following step-level loss:

$$\mathcal{L}^{\text{step}}_{\pi_\theta}(\theta) = \mathbb{E}[p \sim \mathcal{D}_{step}, \{s'_i\}^G_{i=1} \sim \pi_\theta(S|p)] \frac{1}{G} \sum_{i=1}^{G} \Big( \frac{1}{\sum_{t=1}^{|s'_i|} I(s^t_{i,t})} \sum_{t:I(s'_{i,t})=1}^{|s'_i|}$$
$$\min \Big( \frac{\pi_\theta(s'_i|p)}{\pi_{\theta_{\text{old}}}(s'_i|p)} A'_i, \text{clip}(\frac{\pi_\theta(s'_i|p)}{\pi_{\theta_{\text{old}}}(s'_i|p)}, 1 - \varepsilon_{\text{low}}, 1 + \varepsilon_{\text{high}}) A'_i) \Big) + \alpha \mathcal{L}_{\text{NLL}}(\theta). \tag{8}$$

where $s'_i$ and $A'_i$ denote the responses and advantages at the step-level. In step-level optimization, each sample contains only a single think–act–observe loop, and its reward is directly determined by whether the corresponding code execution succeeds. For a group of step-level rollouts sampled from the same prefix $\{s'_i\}^G_{i=1}$, the advantage is computed as:

$$A'_i = \frac{r'_i - \text{mean}(r')}{\text{std}(r')} \tag{9}$$

where $r'$ denotes the code-execution reward within the group. The final training objective combines both episode-level and step-level losses:

$$\mathcal{L}(\theta) = \mathcal{L}^{\text{epis}}_{\pi_\theta}(\theta) + \mathcal{L}^{\text{step}}_{\pi_\theta}(\theta). \tag{10}$$

## 2.4 SELF-CORRECTION DURING INFERENCE

During inference, our model follows the think-act-observe loop. To exploit immediate feedback from tool execution, we introduce a self-correction mechanism that dynamically corrects erroneous reasoning steps, as shown in Figure 3(c). Specifically, when an action $a^t$ fails to execute, it indicates that both the action $a^t$ and its associated reasoning step $r^t$ are likely incorrect. To explore alternative solving paths, the model backtracks to $r^t$ and partitions it into a prefix $r^t_{\text{pre}}$ and a suffix $r^t_{\text{suf}}$, as

Table 1: Comparison with state-of-the-art methods on mathematical benchmarks, the best results are in **bold** and the second-best are underlined. Code use indicates whether code tools are employed. † denotes our reimplementation results of Avg@4, ‡ indicates results from their official releases.

| Model | Code Use | MATH 500 | AIME 2024 | AIME 2025 | AMC 2023 | Minerva Math | Olympiad Bench | Avg. |
|---|---|---|---|---|---|---|---|---|
| *Non-Reasoning Models (Lightweight)* | | | | | | | | |
| Qwen3-1.7B † | | 72.0 | 15.8 | 7.5 | 41.9 | **39.0** | 42.2 | 36.4 |
| Qwen2.5-Math-1.5B † | ✓ | 40.0 | 7.1 | 5.4 | 22.7 | 15.7 | 19.7 | 18.4 |
| THOR-1.5B | ✓ | **78.2** | **36.7** | **20.0** | **67.5** | 38.2 | **54.0** | **49.1** |
| *Non-Reasoning Models (Standard-scale)* | | | | | | | | |
| GPT-4o-1120 † | | 77.2 | 11.1 | 7.6 | 60.0 | 53.4 | 43.9 | 42.2 |
| rStar-Math-7B ‡ | | 78.4 | 26.7 | - | 47.5 | - | 47.1 | - |
| Eurus-2-PRIME-7B ‡ | | 79.2 | 26.7 | 13.3 | 57.4 | 38.6 | 42.1 | 42.9 |
| ARTIST-7B ‡ | ✓ | 67.6 | 15.6 | - | 47.0 | - | 37.9 | - |
| TATA-7B ‡ | ✓ | 73.0 | - | - | - | - | 35.9 | - |
| TORL-7B † | ✓ | 82.2 | 40.8 | 29.2 | 73.8 | 53.8 | 57.3 | 56.2 |
| AutoTIR-7B ‡ | ✓ | 62.6 | 33.3 | 16.7 | - | - | - | - |
| ZTRL-7B ‡ | ✓ | 80.2 | **50.0** | 26.7 | - | - | - | - |
| Qwen2.5-Math-7B-Instruct † | | 79.8 | 10.8 | 11.7 | 54.4 | 44.8 | 43.1 | 40.9 |
| Qwen2.5-Math-7B † | | 51.5 | 8.3 | 5.8 | 33.1 | 26.7 | 22.9 | 24.7 |
| Qwen2.5-Math-7B † | ✓ | 64.7 | 20.6 | 13.1 | 49.7 | 28.1 | 37.9 | 35.7 |
| THOR-7B | ✓ | **87.5** | **50.0** | **33.3** | **81.3** | **53.9** | **61.1** | **61.2** |
| *Reasoning Models (Lightweight)* | | | | | | | | |
| DeepSeek-R1-Distill-Qwen-1.5B ‡ | | 82.8 | 28.9 | 23.3 | 62.9 | 26.5 | 43.3 | 44.6 |
| DeepScaleR-1.5B-Preview ‡ | | 87.8 | 43.1 | 30.0 | 73.6 | 30.2 | 50.0 | 52.5 |
| Qwen3-1.7B † | | 91.0 | 45.0 | 31.7 | 80.6 | 52.7 | 65.7 | 61.1 |
| THOR-Thinking-1.7B | ✓ | **92.8** | **60.0** | **33.3** | **82.5** | **54.4** | **68.8** | **65.3** |
| *Reasoning Models (Standard-scale)* | | | | | | | | |
| QwQ-32B-Preview ‡ | | 90.6 | 50.0 | 40.0 | 80.0 | 39.0 | 58.5 | 59.7 |
| START-32B ‡ | ✓ | 94.4 | 66.7 | 47.1 | 95.0 | - | - | - |
| OpenMath-Nemotron-7B ‡ | ✓ | - | 72.9 | 57.5 | - | - | - | - |
| DeepSeek-R1-Distill-Qwen-7B † | | 93.5 | 55.8 | 40.0 | 86.8 | 57.7 | 71.0 | 67.5 |
| Qwen3-8B † | | 95.5 | 64.2 | 54.2 | 91.2 | 64.4 | 77.7 | 74.5 |
| THOR-Thinking-8B | ✓ | **96.8** | **77.5** | **62.5** | **96.8** | **65.6** | **79.7** | **79.8** |

previously described. Conditioned on the history up to $r_{\text{pre}}^t$, the model then regenerates a new reasoning suffix $\hat{r}_{\text{suf}}^t$ and a revised action $\hat{a}^t$, thereby enabling online error correction during inference. The correction procedure can be repeated for up to $N_{\text{corr}}$ attempts. Importantly, each attempt only requires regenerating the suffix $\hat{r}_{\text{suf}}^t$ and the corresponding action $\hat{a}^t$, rather than recomputing the entire trajectory. Thus, the additional computational cost is minimal compared to the total cost.

## 3 EXPERIMENTS

### 3.1 DATASET

We evaluate the effectiveness of THOR on a diverse set of representative and challenging benchmarks for both mathematical reasoning and code generation. For mathematical reasoning, our evaluation covers the high school-level MATH 500 (Hendrycks et al., 2021), as well as competition-level benchmarks including AMC 2023, AIME 2024, AIME 2025, MinervaMath (Lewkowycz et al., 2022), and OlympiadBench (He et al., 2024). These benchmarks span geometry, algebra, and number theory. To evaluate answer correctness, we adopt the LLM-as-Judge approach, using Qwen3-32B as the judge model to compare model predictions against the ground truth. For code generation, we adopt EvalPlus (Liu et al., 2023) on HumanEval[+] and MBPP[+] to assess basic programming skills, and LiveCodeBench[v6] (Jain et al., 2024) for competition-level programming tasks.

### 3.2 COMPARISON WITH STATE-OF-THE-ART METHODS

To assess THOR's effectiveness and generalization, we conduct extensive experiments on both non-reasoning and reasoning models. For the non-reasoning setting, we use Qwen2.5-Math-7B (Yang

Table 2: Comparison of test-time scaling methods under an equal compute budget ($N = 8$). The proposed self-rewarded strategy relies solely on execution pass rate as an intrinsic signal and does not use any external PRMs.

| Model | MATH 500 | AIME 2024 | AIME 2025 | AMC 2023 | Minerva Math | Olympiad Bench | Avg. |
|---|---|---|---|---|---|---|---|
| *Non-Reasoning Model* | | | | | | | |
| THOR-7B | 87.5 | 50.0 | 33.3 | 81.3 | 53.9 | 61.1 | 61.2 |
|   + Self-Consistency | 89.8 | 50.0 | 36.7 | 80.0 | 56.3 | 64.8 | 62.9 |
|   + InternLM2-RM | 88.2 | 56.7 | 36.7 | 77.5 | 55.9 | 62.8 | 63.0 |
|   + Qwen2.5-Math-PRM | 89.6 | 53.3 | 33.3 | 77.5 | 57.7 | 66.5 | 63.0 |
|   + Self-rewarded | 87.7 | 53.3 | 38.3 | 83.8 | 54.9 | 61.5 | $63.3^{\uparrow 2.1\%}$ |
|   + InternLM2-RM + Self-rewarded | 88.0 | 60.0 | 33.3 | 85.0 | 56.3 | 63.5 | $64.4^{\uparrow 3.2\%}$ |
| *Reasoning Model* | | | | | | | |
| THOR-Thinking-8B | 96.8 | 77.5 | 62.5 | 96.8 | 65.6 | 79.7 | 79.8 |
|   + Self-Consistency | 97.4 | 90.0 | 66.7 | 95.0 | 63.6 | 81.6 | 82.4 |
|   + InternLM2-RM | 96.8 | 83.3 | 73.3 | 97.5 | 66.5 | 81.5 | 83.2 |
|   + Qwen2.5-Math-PRM | 97.0 | 73.3 | 63.3 | 95.0 | 64.7 | 80.4 | 79.0 |
|   + Self-rewarded | 97.2 | 86.7 | 70.0 | 97.5 | 65.8 | 82.0 | $83.2^{\uparrow 3.4\%}$ |
|   + InternLM2-RM + Self-rewarded | 98.0 | 86.7 | 73.3 | 100.0 | 65.4 | 82.1 | $84.3^{\uparrow 4.5\%}$ |

et al., 2024) to obtain THOR-7B. For the reasoning setting, we adopt Qwen3-8B (Thinking Mode) (Yang et al., 2025) to obtain THOR-Thinking-8B. We further test generalization on the corresponding lightweight models, Qwen2.5-Math-1.5B and Qwen3-1.7B. To mitigate randomness, we adopt Avg@4 as the evaluation metric. The maximum context length is 16,384 tokens for reasoning models and 4,096 tokens for non-reasoning models. See Section D for implementation details.

For comparison, we evaluate THOR against a diverse set of TIR and CoT-based methods, including ARTIST-7B (Singh et al., 2025), TATA-7B (Xu et al., 2025), AutoTIR (Wei et al., 2025), TORL-7B (Li et al., 2025b), Eurus-2-PRIME-7B (Cui et al., 2025), rStart-Math-7B (Guan et al., 2025), ZTRL-7B (Mai et al., 2025), and GPT-4o (Hurst et al., 2024). We also include long CoT methods include START (Li et al., 2025a), DeepSeek-R1-Distill-Qwen (Guo et al., 2025), DeepScaleR (Luo et al., 2025), QwQ (Team, 2024) and Nemotron (Moshkov et al., 2025). As shown in Table 1, THOR achieves substantial improvements on both non-reasoning and reasoning models, demonstrating its effectiveness in enhancing mathematical reasoning capabilities. Moreover, despite relying only on small policy models, THOR remains competitive with state-of-the-art systems and even surpasses many larger models, while maintaining low inference cost. Detailed inference costs are provided in Appendix C.2.3.

## 3.3 SELF-REWARDED INFERENCE ENHANCEMENT

Traditional test-time scaling computation approaches, such as Best-of-N (BoN), often rely on external Outcome Reward Models (ORMs) to evaluate the trajectory quality. We propose an ORM-free search method that exploits intermediate code execution feedback as an intrinsic reward signal. Specifically, we sample $N$ independent candidates and select the one with the highest execution pass rate, thereby removing the need for an external reward model. Each candidate does not employ the self-correction mechanism. As shown in Table 2, the self-rewarded BoN strategy significantly outperforms single path reasoning by exploring a larger search space, indicating that code execution success can serve as a reliable reward signal for assessing reasoning quality. Notably, the performance gains are more pronounced on challenging datasets such as AIME 2024 and 2025, indicating that complex reasoning tasks benefit more from precise code execution.

We compare our self-rewarded selection with self-consistency (Chen et al., 2023) and external PRMs, including InternLM2-1.8B-RM (Cai et al., 2024) and Qwen2.5-Math-PRM (Zhang et al., 2025). We further evaluate a hybrid strategy that combines PRMs with our intrinsic signal. When $N$ is large, multiple candidates often share identical code-pass rates; in such cases, PRMs provide an additional preference score that helps separate candidates with the same score. As shown in Table 2, the self-rewarded inference method already matches or even surpasses PRM-based selection.

Table 3: Results of the ablation on each component. Cold start uses the data generated by TIRGen in Section 2.2. EpisRL and StepRL correspond to episode-level and step-level optimization defined in Section 2.3. SelfCorr denotes self-correction during inference in Section 2.4.

| | Code Use | Cold Start | Epis RL | Step RL | Self Corr | MATH 500 | AIME 2024 | AIME 2025 | AMC 2023 | Minerva Math | Olympiad Bench | Avg. |
|---|---|---|---|---|---|---|---|---|---|---|---|---|
| *Non-Reasoning Model* | | | | | | | | | | | | |
| T1 | | | | | | 51.5 | 8.3 | 5.8 | 33.1 | 26.7 | 22.9 | 24.7 |
| T2 | | | ✓ | | | 72.9 | 30.0 | 11.7 | 53.8 | 41.5 | 38.3 | 41.4 |
| T3 | ✓ | ✓ | | | | 64.7 | 20.6 | 13.3 | 49.7 | 28.1 | 37.9 | 35.7 |
| T4 | ✓ | ✓ | ✓ | | | 86.9 | 42.7 | 30.8 | 77.5 | 52.2 | 58.2 | 58.1 |
| T5 | ✓ | ✓ | ✓ | ✓ | | 87.3 | 45.0 | 31.7 | 80.0 | 53.4 | 60.5 | 59.7 |
| T6 | ✓ | ✓ | ✓ | ✓ | ✓ | 87.5 | 50.0 | 33.3 | 81.3 | 53.9 | 61.1 | 61.2 |
| *Reasoning Model* | | | | | | | | | | | | |
| T1 | | | | | | 95.5 | 64.2 | 54.2 | 91.2 | 64.4 | 77.7 | 74.5 |
| T2 | | | ✓ | | | 95.7 | 65.8 | 52.5 | 93.1 | 64.4 | 78.0 | 74.9 |
| T3 | ✓ | ✓ | | | | 92.9 | 60.8 | 51.7 | 88.8 | 61.4 | 72.9 | 71.4 |
| T4 | ✓ | ✓ | ✓ | | | 96.1 | 71.7 | 60.0 | 95.0 | 64.5 | 78.9 | 77.7 |
| T5 | ✓ | ✓ | ✓ | ✓ | | 96.6 | 74.2 | 60.0 | 95.6 | 65.4 | 79.0 | 78.5 |
| T6 | ✓ | ✓ | ✓ | ✓ | ✓ | 96.8 | 77.5 | 62.5 | 96.8 | 65.6 | 79.7 | 79.8 |

Moreover, combining it with PRMs yields further improvements, indicating that the two types of signals are complementary. In addition, our reward is model-agnostic and benefits both reasoning and non-reasoning models, whereas PRMs are fundamentally limited by the coverage and biases of their training data, which often restricts generalization.

## 3.4 ABLATION STUDY

To analyze the contribution of each component in THOR, we conduct an ablation study by selectively removing key modules. This results in six system variants T1–T6, built upon Qwen2.5-Math-7B and Qwen3-8B, as summarized in Table 3.

**Impact of Cold Start.** The cold start data generated by TIRGen provides a foundation for subsequent RL. The goal of RL is to refine the model's policy towards its capability frontier (Yue et al., 2025a), which can be estimated using pass@k (Chen et al., 2021). Consequently, we evaluate cold start by its impact on pass@16 and code invocation ratio. We further compare with other TIR dataset, including the Long CoT TIR data generated by Nemotron (Moshkov et al., 2025) and the Short CoT TIR data from ReTool (Feng et al., 2025). As shown in Figure 4, TIRGen substantially improves both metrics, effectively expanding the capability frontier. Compared with other datasets, the data generated by TIRGen effectively mitigates performance degradation arising from out-of-distribution samples. More critically, it actively encourages reasoning models to utilize tools and dramatically increases the code ratio, a behavior rarely seen in the baseline.

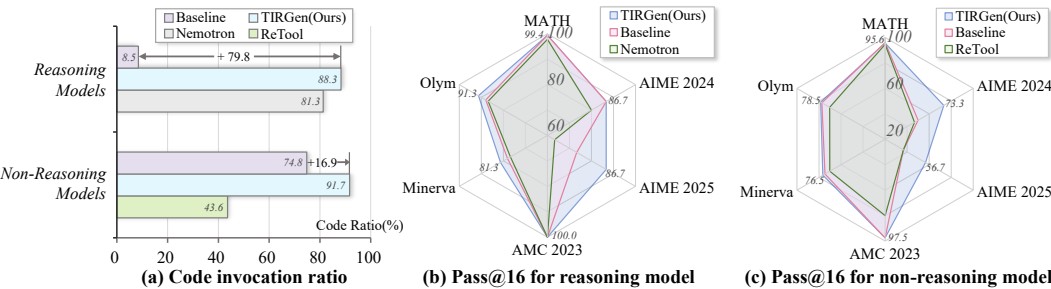

Figure 4: Ablation on cold-start efficiency. We compare our TIRGen against other TIR datasets, including Long CoT from Nemotron and Short CoT from ReTool. Results are reported as code ratio in (a) and pass@16 in (b) and (c), demonstrating the effectiveness of TIRGen and cold start.

**Impact of Tool-Integrated RL.** To evaluate the effectiveness of tool-integrated RL, we apply two different RL strategies to the baseline model (T1): a standard CoT-based RL and a episode-level TIR-based RL, which yield T2 and T4, respectively. While both outperform the baseline, T4 achieves substantially greater improvements than T2, validating the effectiveness of TIR RL.

**Impact of Hierarchical RL.** By incorporating step-level optimization into episode-level RL (T4), we obtain T5. T5 achieves further performance gains across most datasets, underscoring the importance of fine-grained optimization for enhancing code generation capabilities in a TIR setting.

**Impact of Self-Correction.** By leveraging step-level feedback from intermediate code, we construct a self-correction mechanism, yielding variant T6. We set the maximum number of correction attempts $N_{corr} = 4$, which leads to substantial performance gains. This result highlights the critical importance of successful code generation and execution for the final outcome.

## 3.5 GENERALIZATION ON CODE BENCHMARKS

We also evaluate THOR's code generation abilities using the pass@1 metric on HumanEval$^+$, MBPP$^+$ and LiveCodeBench$^{v6}$. As illustrated in Figure 5, our models achieve consistent improvements across all benchmarks. Notably, these gains are realized in a zero-shot setting without any fine-tuning on code generation data. These results confirm that our method strengthens both mathematical reasoning and code generation, underscoring THOR's robustness and versatility across distinct reasoning domains.

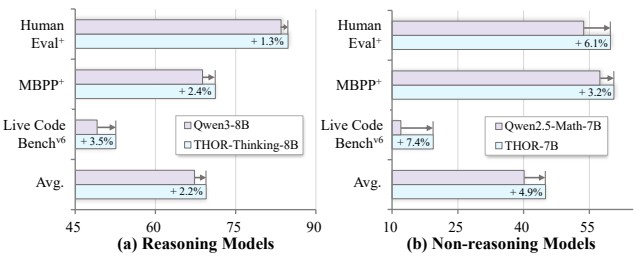

Figure 5: Pass@1 accuracy on code generation benchmarks.

## 4 CONCLUSION

In this work, we address three core challenges in tool-integrated reasoning: construction of TIR data, hierarchical optimization, and inference enhancement. We propose THOR (**T**ool-Integrated **H**ierarchical **O**ptimization via **R**L), a novel hierarchical RL framework for TIR that fully leverages step-level feedback. First, to mitigate the scarcity of TIR data, we introduce TIRGen, an multi-agent-based TIR data construction pipeline. For model training, THOR integrates coarse-grained episode-level optimization for overall reasoning ability with fine-grained step-level optimization for code generation ability. During inference, tool feedback is used to dynamically adjust the reasoning and perform self-correction. Experiments demonstrate that THOR achieves substantial improvements across diverse models and benchmarks while maintaining a low inference cost.

### REPRODUCIBILITY STATEMENT

For the reproducibility of our results, we have provided a detailed description of our method in Section 2 and experimental setups in Appendix D. In addition, to further facilitate the reproduction, we will release our codes and datasets.

### ETHICS STATEMENT

By integrating precise tool execution with hierarchical reinforcement learning, THOR significantly enhances the mathematical reasoning capabilities of LLMs. This advancement holds substantial promise for education and scientific research by offering reliable, automated assistance for complex problems in mathematics, engineering, and the formal sciences. However, like any powerful LLM-based system, THOR carries a risk of misuse, such as generating misleading or harmful content if deployed without proper oversight. Consequently, the development of robust ethical safeguards and responsible deployment protocols is important for its application in real-world scenarios.

ACKNOWLEDGMENTS

This work was supported by the National Natural Science Foundation of China under Grant No. U25A20409.

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

## A  LLM USAGE

We used LLMs, including GPT-5 and Gemini 2.5 Pro, only to polish grammar and improve the clarity of the manuscript. All research ideas, experiments, and analyses were conducted by the authors.

## B  RELATED WORKS

LLMs have shown remarkable progress in mathematical problem solving. This section reviews relevant works, which we categorize into two main groups based on whether they integrate external tools: tool-free reasoning and tool-integrated reasoning.

### B.1  TOOL-FREE REASONING

**Search-based Methods**. Search-based methods improve LLM reasoning by systematically exploring a large space of potential solutions. Early approaches leveraged prompting strategies like Chain-of-Thought (CoT) (Wei et al., 2022) and its search-oriented extensions, including Tree-of-Thought (ToT) (Yao et al., 2023a) and Graph-of-Thought (GoT) (Besta et al., 2024). To more effectively guide this exploration, a prominent line of work integrates explicit Reward Models (RMs), including Outcome Reward Models (ORMs) (Yu et al., 2023; Zhang et al., 2024) and Process Reward Models (PRMs) (Wang et al., 2023b; 2024; Zhang et al., 2025) with search algorithms like BoN, step-level BoN, MCTS, and beam search. For instance, Marco-o1 (Zhao et al., 2024) and rStar-Math (Guan et al., 2025) employ MCTS for systematic exploration, while PRM-BAS (Hu et al., 2025) uses Beam Annealing Search to balance search breadth and efficiency. Although these methods yield significant performance gains, they have two key limitations. First, the large-scale search incurs substantial computational overhead at inference time. Second, and more critically, they do not directly optimize the model's internal reasoning policy, thereby constraining its ultimate capability.

**RL-based Methods**. RL represents another dominant paradigm for enhancing LLM reasoning, where policy gradient (PG) methods have become a core technical route (Guo et al., 2025; Team et al., 2025; Yang et al., 2025). These methods can be categorized into value-model-based and value-model-free approaches. Value-model-based approaches are exemplified by Proximal Policy Optimization (PPO) (Schulman et al., 2017), which stabilizes training via policy probability ratio clipping. Its variants include VC-PPO (Yuan et al., 2025) with a decoupled-GAE to mitigate value bias and reward decay, and VAPO (Yue et al., 2025b) with a length-adaptive GAE to address the bias-variance trade-off. Value-model-free methods bypass the need for an explicit value critic. For instance, GRPO (Shao et al., 2024) estimates the baseline from group scores. This is enhanced by DAPO (Yu et al., 2025) with techniques like dynamic sampling and token-level loss, while GSPO (Zheng et al., 2025) performs optimization based on sequence-level likelihood ratios. A key limitation of these approaches is their primary reliance on sparse, episode-level reward signals. For tasks involving long reasoning chains, this reward sparsity impedes efficient policy learning.

### B.2  TOOL-INTEGRATED REASONING

Integrating external tools, such as code executors, search engines, databases, and external APIs, has become a prominent strategy for augmenting the reasoning capabilities of LLMs while enhancing their interpretability (Liao et al., 2026). Early approaches focused on prompting methods, without integrating tool use into model optimization. For example, ReAct (Yao et al., 2023b) demonstrated the use of prompting to invoke the Wikipedia API for question answering and fact verification. VOYAGER (Wang et al., 2023a) explored in-context learning to leverage predefined tools within Minecraft. Subsequent studies incorporated human-labeled or synthetic tool-integrated data during fine-tuning (Schick et al., 2023; Yang et al., 2024; Moshkov et al., 2025; Li et al., 2025a). However, while effective in specific domains, the generalization is often constrained by the scope and quality of the synthetic data. More recently, RL has been employed to learn dynamic tool-integrated policies for mathematics reasoning (Mai et al., 2025; Li et al., 2025b; Feng et al., 2025) and other tasks (Qian et al., 2025). Nevertheless, existing RL-based approaches often rely on prompt-based triggers to initiate tool invocation, which limits their applicability to models not previously exposed

to tool-integrated training data, such as DeepSeek-R1 (Guo et al., 2025) and QwQ (Team, 2025). Furthermore, the step-level feedback provided by tools remains unexplored.

## C  MORE EXPERIMENTS

### C.1  STATISTICAL VALIDATION

To examine the relationship between code execution success and answer correctness, we analyzed their joint distribution on the test set, as shown in Table 4. We then applied a chi-square test of independence, which yielded a highly significant result ($\chi^2 = 336.3, p = 4.09 \times 10^{-75}$), thereby rejecting the null hypothesis of independence and confirming a statistical association between the two variables. These findings verified our research motivation: *the success of an intermediate tool call is a strong predictor of the final answer's correctness*.

Table 4: Joint distribution between code execution result and answer correctness.

|  | Code True | Code False |
|---|---|---|
| Answer True | 3950 | 139 |
| Answer False | 1549 | 318 |

### C.2  MORE ANALYSIS ON SELF-CORRECTION AND INFERENCE COST

#### C.2.1  ABLATION ON EXECUTION FAILURE REPAIR STRATEGIES

In Section 2.4, we introduced a self-correction mechanism that backtracks to the most recent reasoning step and regenerates a suffix when code execution fails. While intuitive, execution failures can arise from different sources, including (i) syntax/runtime errors within the generated code and (ii) deeper logical inconsistencies in the preceding reasoning step. To disentangle these cases and evaluate the necessity of suffix-level regeneration, we conduct a controlled ablation in Table 5, comparing three alternative repair strategies: (a) Action-only repair, (b) Step-suffix repair, (c) Full re-repair.

Table 5: Ablation of different repair strategies for self-correction during inference.

| Model | MATH 500 | AIME 2024 | AIME 2025 | AMC 2023 | Minerva Math | Olympiad Bench | Avg. |
|---|---|---|---|---|---|---|---|
| *Non-Reasoning Model* | | | | | | | |
| THOR-7B | 87.3 | 45.0 | 31.7 | 80.0 | 53.4 | 60.5 | 59.7 |
| THOR-7B + action-only repair | 87.0 | 46.3 | 32.1 | 82.5 | 53.3 | 60.6 | 60.3 |
| THOR-7B + step-suffix repair | 87.5 | 50.0 | 33.3 | 81.3 | 53.9 | 61.1 | **61.2** |
| THOR-7B + full re-plan | 86.6 | 46.7 | 34.2 | 81.3 | 52.2 | 61.1 | 60.4 |
| *Reasoning Model* | | | | | | | |
| THOR-Thinking-8B | 96.6 | 74.2 | 60.0 | 95.6 | 65.4 | 79.0 | 78.5 |
| THOR-Thinking-8B + action-only repair | 96.8 | 75.0 | 60.0 | 95.6 | 64.0 | 79.7 | 78.5 |
| THOR-Thinking-8B + step-suffix repair | 96.8 | 77.5 | 62.5 | 96.8 | 65.6 | 79.7 | **79.8** |
| THOR-Thinking-8B + full re-plan | 96.2 | 75.8 | 62.5 | 98.1 | 63.4 | 79.3 | 79.2 |

#### C.2.2  ANALYSIS OF SELF-CORRECTION MECHANISMS

To further evaluate the effect of explicit self-correction, we compare three settings: the baseline model without any correction, THOR-7B relying solely on its emergent self-correction behavior, and THOR-7B equipped with our explicit self-correction mechanism. We report results across accuracy, code ratio, the number of failed and successful tool calls, and the overall code pass rate. As shown in Table 6, the explicit mechanism markedly reduces failed executions, significantly increases the code pass rate, and yields consistent improvements in accuracy.

Table 6: Comparison of emergent and explicit self-correction across accuracy and code pass metrics.

| Model | Acc | Code Ratio | Failed Code Num | Success Code Num | Code Pass Ratio |
|---|---|---|---|---|---|
| *Non-Reasoning Model* | | | | | |
| Qwen2.5-Math-7B | 35.7 | 72.7 | 858 | 5806 | 87.1 |
| THOR-7B | 59.7 | 96.6 | 685 | 5900 | 89.6 |
| THOR-7B + self-correction | 61.2 | 96.2 | 82 | 6050 | 98.7 |
| *Reasoning Model* | | | | | |
| Qwen3-8B | 74.5 | 8.5 | 260 | 1071 | 80.5 |
| THOR-Thinking-8B | 78.5 | 91.3 | 2311 | 10535 | 82.0 |
| THOR-Thinking-8B + self-correction | 79.8 | 90.6 | 155 | 12532 | 99.0 |

### C.2.3 INFERENCE COST ANALYSIS

We evaluate the inference efficiency of THOR by analyzing its token consumption and runtime performance. Our data construction process, guided by TIRGen, identifies redundant computational steps within reasoning chains and transforms them into executable code. By learning from this data, THOR is trained to generate more concise solutions, effectively leveraging tools to simplify computations at inference time.

In addition to token usage, we further evaluate the total inference seconds and frames-per-second (FPS) throughput under a unified vLLM inference environment. As reported in Table 7, THOR consistently generates fewer tokens than the baseline models, while also reducing overall inference latency. These reductions directly translate into improved FPS. These results already include the overhead of self-correction.

Table 7: The number of tokens consumed during inference across different benchmarks.

| Model | MATH 500 | AIME 2024 | AIME 2025 | AMC 2023 | Minerva Math | Olympiad Bench | Avg Tokens | Infer Seconds | FPS |
|---|---|---|---|---|---|---|---|---|---|
| *Non-Reasoning Model* | | | | | | | | | |
| Qwen2.5-Math-7B | 866 | 1283 | 1325 | 1132 | 802 | 1090 | 1083 | 1379 | 4.48 |
| THOR-7B | 705 | 1351 | 1420 | 928 | 729 | 981 | $1019^{\downarrow 6\%}$ | 1324 | $4.67^{\uparrow 4\%}$ |
| *Reasoning Model* | | | | | | | | | |
| Qwen3-8B | 5102 | 11986 | 13022 | 7989 | 6906 | 9238 | 9041 | 7244 | 0.85 |
| THOR-Thinking-8B | 4506 | 10338 | 11807 | 6749 | 5444 | 8205 | $7841^{\downarrow 13\%}$ | 6280 | $0.98^{\uparrow 15\%}$ |

## D IMPLEMENTATION DETAILS

### D.1 TIR DATA CONSTRUCTION & COLD START

In the cold start stage, for data source construction, we collected a large set of question-answer pairs from various public datasets, including DAPO17k (Yu et al., 2025), DeepMath103k (He et al., 2025), and Deepscaler40k (Luo et al., 2025), which cover mathematical problems of diverse difficulty levels. For sampling, we set the hyperparameters to a temperature of 0.6, top-$k$ of 50, and top-$p$ of 1.0. After processing with TIRGen, we obtain 29,217 short CoT TIR samples from Qwen2.5-Math-7B and 57,598 long CoT TIR samples from Qwen3-8B. The distribution of code invocation counts in the final cold start dataset $\mathcal{D}_{SFT}$ is shown in Figure 6. In TIRGen, the Generator agent uses the corresponding policy model, while the Refiner agent uses Qwen3-32B (Non-thinking) for its strong instruction-following capability. For non-reasoning models, we set $L_{\text{step}} = 512$, whereas for reasoning models we set $L_{\text{step}} = 4096$. Our experiments utilize SandboxFusion[1] as the external code execution environment.

For cold start, models are full-parameter fine-tuned for 1 epoch with a global batch size of 256. We use AdamW optimizer (Loshchilov & Hutter, 2017) with a fixed learning rate of $2 \times 10^{-6}$.

---

[1]https://github.com/bytedance/SandboxFusion

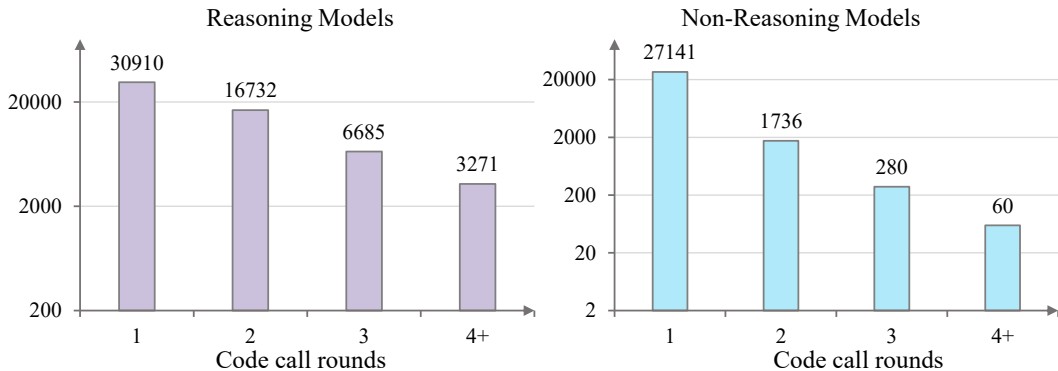

Figure 6: The distribution of code call rounds in the cold start dataset $\mathcal{D}_{SFT}$.

ZeRO (Rajbhandari et al., 2020) is adopted for memory-efficient training. For reasoning models, the maximum context length is 20,000 tokens, while for non-reasoning models it is 4,096 tokens.

### D.2 HIERARCHICAL REINFORCEMENT LEARNING

During the RL stage, we use the publicly available dataset ToRL28k (Li et al., 2025b). To stabilize training, we adopt an off-policy variant of GRPO and employ dynamic data filtering (Yu et al., 2025) to accelerate convergence. In the early phase of step-level optimization, the code execution success rate is relatively low, resulting in many failed steps. We downsample these failed steps to ensure that the number of step-level optimization queries does not exceed that of episode-level optimization, thereby prioritizing the model's overall problem-solving ability.

We set the group size $G = 16$, the group sampling temperature to $1.0$, the weight coefficient on the loss $\mathcal{L}_{\text{NLL}}$ to $\alpha = 0.01$, and the learning rate to $1 \times 10^{-6}$. The KL-divergence term is omitted. Clipping coefficients are configured as $\varepsilon_{\text{high}} = 0.28$ to encourage diversity and $\varepsilon_{\text{low}} = 0.2$. During rollout, we use the maximum sampling lengths of 4,096 tokens for non-reasoning models and 16,384 tokens for reasoning models, with up to 5 rounds of code interaction. In the step-level optimization, the backtracking length $L_{\text{suf}}$ is set to 500 for reasoning models and 100 for non-reasoning models. A rule-based reward function is used to mitigate reward hacking. All experiments are conducted on 16 NVIDIA H200 GPUs.

### D.3 INFERENCE WITH SELF-CORRECTION

During inference, we set the sampling parameters to a temperature of $0.6$, top-$p$ of $1.0$, and top-$k$ of 50. The maximum number of self-correction attempts $N_{\text{corr}}$ is set to 4. For non-reasoning models, the backtracking length $L_{\text{suf}}$ is set to 100, while for reasoning models it is set to 500.

### D.4 PROMPT SETTING

In this subsection, we provide the complete prompt settings used in our framework. Figures 7 and 8 illustrate the prompts designed for the Generator and Refiner agents in the TIRGen data construction pipeline, respectively. Figures 9 and 10 present the prompts for tool-integrated reasoning in reasoning models and non-reasoning models.

## E LIMITATION AND FUTURE WORK

In this work, we systematically investigate the effectiveness of tool-integrated reasoning, focusing specifically on code integration for mathematical problem solving. Although we have verified the effectiveness of code-integrated reasoning, other types of tools, such as search engines, symbolic systems remain to be explored. Due to computational constraints, we did not experiment with larger-scale models such as 32B or 72B. Nevertheless, we have validated the effectiveness of THOR across

multiple model sizes ranging from 1.5B to 8B, which demonstrates that our approach generalizes well across different scales. In the future, we will explore larger models and conduct a deeper investigation into multi-tool joint optimization.

## F    CASE STUDY

In this section, we present a case study to illustrate how THOR performs tool-integrated reasoning, including non-reasoning models in Figure 11, 12, and reasoning models in Figure 13.

You are a scientist proficient in computer science and mathematics. I will provide you with a detailed thought process (chain of thought) from a powerful model for a mathematical problem. Your task is to revise this thought process.

## Revision Goal:
Without altering the original model's reasoning flow and methods, identify any steps in the chain of thought that can be assisted by code for numerical calculations, equation solving, hypothesis testing, data processing, etc., and replace these natural language descriptions of computational processes with corresponding code execution and predicted results.

### You need to complete the following steps:
1. Identify Codable Parts: Carefully analyze the original chain of thought to pinpoint any parts involving specific numerical calculations, algebraic operations, set operations, logical verifications, etc., that can be precisely executed using Python code.
2. Write Python Code: For the identified parts, write clear, concise Python code blocks that can accomplish the corresponding computational tasks. Each piece of your code should be carefully considered, not just performing simple arithmetic operations like addition, subtraction, multiplication, or division. Every code block should return the output with `print()` function.
3. Predict Code Output: Provide the expected output of the Python code you have written.
4. Embed in Chain of Thought: Embed the Python code and its execution results into the chain of thought, ensuring that the revised thought process remains logically correct and complete. Keep the parts that were not modified exactly as they were! Also, only modify the parts included in the original chain of thought; do not extend or continue solving parts not covered by the chain of thought.

### Note:
1. Multiple Code Blocks Supported: You can output multiple code blocks as needed to assist different computational steps in the reasoning process.
2. Independent Code Blocks: Each of your code blocks is independent, can run on its own, does not depend on any previous variables, and imports any required libraries independently.
3. Maintain Consistency of Thought: Do not alter the core reasoning logic, step order, or basic methods of the original chain of thought. Your task is solely to convert specific computational and verification processes described in natural language into equivalent code execution and output.
4. Accuracy: Ensure that the code you provide is correct and that the predicted output is accurate.
5. No Problem Solving: Your task is merely to revise the chain of thought. Parts that do not need modification should remain unchanged and be outputted as such.

If Python code can be used to assist in solving, please strictly follow the format below to ensure your revised content is easily parsable by machines and understandable by humans:

Original reasoning step, …
```python
```
```output
Code output 1
```
Original reasoning steps, …
```python
Python block 2
```
```output
Code output 2
```
Continue reasoning…

Figure 7: The prompt used by the Refiner agent in our TIR data construction pipeline, TIRGen.

Please reason step by step and put your final answer within \\boxed{}. \n\n [question]

Figure 8: The prompt used by the Generator agent in our TIR data construction pipeline, TIRGen.

You are a scientist skilled in mathematics and computer science. Please integrate natural language reasoning and Python code to solve mathematical problems. You can use Python code during the thinking process for numerical calculations, equation solving, hypothesis verification, data processing, and any other steps to assist your reasoning. The final answer should be placed within \\boxed{}.
Note:
1. Independent Code Blocks: Each of your code blocks is independent, can run on its own, does not depend on any previous variables, and imports any required libraries independently.
2. Multiple Code Blocks Supported: You can output multiple code blocks as needed to assist different computational steps in the reasoning process.
3. Careful Thinking: Each piece of code should be carefully considered and go beyond simple arithmetic operations such as addition, subtraction, multiplication, or division. Each code block should include a `print()` function to display its output.

Code Format:
```python
code blocks
```
```output
Code output
```

Question: [question]

Figure 9: Prompt for tool-integrated reasoning in reasoning models.

A conversation between User and Assistant. The user asks a question, and the Assistant solves it. Please integrate natural language reasoning with programs to solve the problem above, and put your final answer within \\boxed{}.

Question: [question]

Figure 10: Prompt for tool-integrated reasoning in non-reasoning models.

**Question:**

Find the largest possible real part of $(75 + 117i)z + \dfrac{96 + 144i}{z}$ where $z$ is a complex number with $|z| = 4$.

**Answer:**

540

**Response:**

Let's solve the problem step by step using Python and SymPy. We need to find the largest possible real part of the expression $(75 + 117i)z + \dfrac{96 + 144i}{z}$ where $z = 4$.

1. Let $z = 4e^{i\theta} = 4(\cos\theta + i\sin\theta)$.
2. Substitute $z$ into the expression and simplify.
3. Find the Real part of the simplified expression.
4. Maximize the real part with respect to $\theta$.

Let's implement this in Python using SymPy.

```python
import sympy as sp
# Define the variable theta
theta = sp.symbols('theta', real=True)
# Define the complex number z
z = 4 * (sp.cos(theta) + sp.I * sp.sin(theta))
# Define the expression
expression = (75 + 117 * sp.I) * z + (96 + 144 * sp.I) / z
# Simplify the expression
simplified_expression = sp.simplify(expression)
# Extract the real part of the expression
real_part = sp.re(simplified_expression)
# Find the maximum value of the real part with respect to theta
max_real_part = sp.maximum(real_part, theta, domain=sp.Interval(0, 2 * sp.pi))
# Output
print(max_real_part)
```

```output
-72/5 - 972*cos(2*atan(4/3))/5 + 2604*sin(2*atan(4/3))/5
```

It seems the output is a bit complicated and not fully simplified. Let's simplify the expression further and find the numerical value of the maximum real part.

We can use SymPy's `evalf` function to get a numerical approximation of the maximum real part. Let's update the code to include this step.

```python
import sympy as sp
# Define the variable theta
theta = sp.symbols('theta', real=True)
# Define the complex number z
z = 4 * (sp.cos(theta) + sp.I * sp.sin(theta))
# Define the expression
expression = (75 + 117 * sp.I) * z + (96 + 144 * sp.I) / z
# Simplify the expression
simplified_expression = sp.simplify(expression)
# Extract the real part of the expression
real_part = sp.re(simplified_expression)
# Find the maximum value of the real part with respect to theta
max_real_part = sp.maximum(real_part, theta, domain=sp.Interval(0, 2 * sp.pi))
# Evaluate the maximum real part numerically
max_real_part_eval = max_real_part.evalf()
print(max_real_part_eval)
```

```output
540.000000000000
```

The largest possible real part of $(75 + 117i)z + \dfrac{96 + 144i}{z}$ where $z$ is a complex number with $|z| = 4$ is $\boxed{540}$.

Figure 11: Example of tool-integrated reasoning by THOR-7B on an AIME 2024 problem, where external tools facilitate precise numerical computation.

**Question:**

The parabola with equation $y = x^2 - 4$ is rotated $60°$ counterclockwise around the origin. The unique point in the fourth quadrant where the original parabola and its image intersect has $y$-coordinate $(a - \sqrt{b})/c$, where a, b, and c are positive integers, and a and c are relatively prime. Find $a + b + c$.

**Answer:**

62

**Response:**

To solve this problem, we need to follow these steps:

1. **Understand the rotation**: We need to rotate the point $(x, y)$ on the parabola $y = x^2 - 4$ by $60°$ counterclockwise around the origin. The rotation matrix for a $60°$ counterclockwise rotation is:

$$\begin{pmatrix} \cos 60° & -\sin 60° \\ \sin 60° & \cos 60° \end{pmatrix} = \begin{pmatrix} \frac{1}{2} & -\frac{\sqrt{3}}{2} \\ \frac{\sqrt{3}}{2} & \frac{1}{2} \end{pmatrix}$$

So, the new coordinates $(x', y')$ after rotation are:

$$\begin{pmatrix} x' \\ y' \end{pmatrix} = \begin{pmatrix} \frac{1}{2} & -\frac{\sqrt{3}}{2} \\ \frac{\sqrt{3}}{2} & \frac{1}{2} \end{pmatrix}\begin{pmatrix} x \\ y \end{pmatrix} = \begin{pmatrix} \frac{x}{2} - \frac{\sqrt{3}y}{2} \\ \frac{\sqrt{3}x}{2} + \frac{y}{2} \end{pmatrix}$$

Since $y = x^2 - 4$, we substitute y in the rotation equations:

$$x' = \frac{x}{2} - \frac{\sqrt{3}(x^2 - 4)}{2}$$

$$y' = \frac{\sqrt{3}x}{2} + \frac{x^2 - 4}{2}$$

2. **Find the intersection**: We need to find the points where the original parabola $y = x^2 - 4$ and its rotated image $y' = (\sqrt{3}x)/2 + (x^2 - 4)/2$ intersect. We set $y = y'$:

$$x^2 - 4 = \frac{\sqrt{3}x}{2} + \frac{x^2 - 4}{2}$$

Simplify the equation:

$$2x^2 - 8 = \sqrt{3}x + x^2 - 4$$

$$x^2 - \sqrt{3}x - 4 = 0$$

Solve this quadratic equation using the quadratic formula $x = (-b \pm \sqrt{b^2 - 4ac})/2a$:

$$x = \frac{\sqrt{3} \pm \sqrt{3 + 16}}{2} = \frac{\sqrt{3} \pm \sqrt{19}}{2}$$

We are looking for the point in the fourth quadrant, so take the negative root:

$$x = \frac{\sqrt{3} - \sqrt{19}}{2}$$

Substitute this x value into the original parabola equation $y = x^2 - 4$:

$$y = \left(\frac{\sqrt{3} - \sqrt{19}}{2}\right)^2 - 4 = \frac{3 - 2\sqrt{57} + 19}{4} - 4 = \frac{22 - 2\sqrt{57}}{4} - 4 = \frac{11 - \sqrt{57}}{2} - 4 = \frac{11 - \sqrt{57} - 8}{2} = \frac{3 - \sqrt{57}}{2}$$

So the y-coordinate is $(3 - \sqrt{57})/2$. Here, a = 3, b = 57, and c = 2. Since a and c are relatively prime, we find a + b + c = 3 + 57 + 2 = 62.

Let's confirm this with Python code to ensure accuracy.

```python
from sympy import symbols, solve, sqrt
# Define the variables
x = symbols('x')
# Original parabola equation
y = x**2 - 4
# Rotated parabola equation
y_prime = (sqrt(3)*x/2) + (x**2 - 4)/2
# Set the original and rotated equations equal to each other
equation = y - y_prime
# Solve for x
x_solutions = solve(equation, x)
# Filter the solution in the fourth quadrant
x_fourth_quadrant = [sol.evalf() for sol in x_solutions if sol.evalf() < 0][0]
# Substitute the x value into the original parabola equation to find y
y_value = y.subs(x, x_fourth_quadrant).evalf()
# Extract a, b, c from the y value
a = 3
b = 57
c = 2
result = a + b + c
print((x_fourth_quadrant, y_value, a, b, c, result))
```

```output
(-1.31342406798590, -2.27491721763537, 3, 57, 2, 62)
```

The x-coordinate of the point in the fourth quadrant is approximately -1.31342406798590. The corresponding y-coordinate is approximately -2.27491721763537. This confirms that the y-coordinate is indeed $(3 - \sqrt{57})/2$. Thus, the values of a, b, and c are 3, 57, and 2 respectively.

Therefore, a + b + c = 3 + 57 + 2 = 62. The final answer is $\boxed{62}$.

Figure 12: Example of tool-integrated reasoning by THOR-7B on an AIME 2025 problem, where complex equations are efficiently solved and verified with tool support.

**Question:**

Let $N$ be the greatest four-digit positive integer with the property that whenever one of its digits is changed to 1, the resulting number is divisible by 7. Let $Q$ and $R$ be the quotient and remainder, respectively, when $N$ is divided by 1000. Find $Q + R$.

**Answer:**

699

**Response:**

<think>

Okay, so I need to find the greatest four-digit number N such that if you change any one of its digits to 1, the resulting number is divisible by 7. Then, once I find N, I have to divide it by 1000 to get Q (the quotient) and R (the remainder), and then add Q and R together. Alright, let me break this down step by step.

=================*More reasoning content are omitted here*=================

To verify this, let me write a Python code snippet to confirm that N=5694 satisfies the conditions.

```python
# Define N
N = 5694
# Check all four conditions
def is_divisible_by_7(num):
    return num % 7 == 0
# Change each digit to 1 and check divisibility
def check_conditions(N):
    # Change first digit to 1
    num1 = 1000 + (N % 1000)
    # Change second digit to 1
    num2 = 1000 * (N // 1000) + 100 + (N % 100)
    # Change third digit to 1
    num3 = 1000 * (N // 1000) + 100 * (N // 100) + 10 + (N % 10)
    # Change fourth digit to 1
    num4 = N - (N % 10) + 1
    return is_divisible_by_7(num1) and is_divisible_by_7(num2) and is_divisible_by_7(num3) and is_divisible_by_7(num4)
# Check if N=5694 satisfies the conditions
print(check_conditions(N))
```

```output
False
```

Wait, the code returns False, which suggests that there is an error in my reasoning. This discrepancy indicates that there's a mistake in the logic or code. Let me debug this.

Let me compute the four numbers manually for N=5694:

=================*More reasoning content are omitted here*=================

Similarly for other digits. Therefore, the code was wrong in the way it was calculating num3. Let me correct the code. Let me rewrite the code:

```python
# Define N
N = 5694
# Check all four conditions
def is_divisible_by_7(num):
    return num % 7 == 0
# Change each digit to 1 and check divisibility
def check_conditions(N):
    # Change first digit to 1
    a = N // 1000
    b = (N // 100) % 10
    c = (N // 10) % 10
    d = N % 10
    num1 = 1000*1 + 100*b + 10*c + d
    # Change second digit to 1
    num2 = 1000*a + 100*1 + 10*c + d
    # Change third digit to 1
    num3 = 1000*a + 100*b + 10*1 + d
    # Change fourth digit to 1
    num4 = 1000*a + 100*b + 10*c + 1
    return is_divisible_by_7(num1) and is_divisible_by_7(num2) and is_divisible_by_7(num3) and is_divisible_by_7(num4)
# Check if N=5694 satisfies the conditions
print(check_conditions(N))
```

```output
True
```

With this corrected code, the function returns `True`, confirming that N=5694 indeed satisfies all the conditions. Therefore, our earlier conclusion was correct, and the mistake was in the code's logic for calculating the numbers when digits are changed. Thus, the final answer is Q + R = 5 + 694 = 699.

**Final Answer:** $\boxed{699}$

</think>

To solve the problem, we are given a four-digit number N such that changing any one of its digits to 1 results in a number divisible by 7. Our goal is to find the **greatest** such number N, and then compute Q + R, where Q is the quotient and R is the remainder when N is divided by 1000.

=================*More reasoning content are omitted here*=================

All conditions are satisfied. Now, divide N = 5694 by 1000:

$Q = \lfloor 5694/1000 \rfloor = 5$, $R = 5694 \bmod 1000 = 694$，$Q + R = 5 + 694 = 699$

**Final Answer:** $\boxed{699}$

Figure 13: Example of tool-integrated reasoning by THOR-Thinking-8B on an AIME 2024 problem, where the use of the tool enables the self-validation of reasoning steps.

