# OpenReview forum: "THOR: Tool-Integrated Hierarchical Optimization via RL for Mathematical Reasoning"
_ICLR.cc/2026/Conference — ICLR 2026 Poster_

### Official Review · Reviewer_RCoA · 2025-10-29

**Soundness:** 3
**Presentation:** 4
**Contribution:** 3
**Rating:** 6
**Confidence:** 4

**Summary:**

THOR is a technically competent and empirically thorough paper that explores hierarchical optimization for tool-integrated reasoning—a topic of rising importance in post-RLHF LLM training. The paper’s main contributions, especially the combination of TIR data generation (TIRGen) and dual-level GRPO optimization, are sound and moderately novel. The experiments are comprehensive, covering both reasoning and code tasks, and the observed gains are consistent.

However, while the framework is methodically interesting, it does not radically depart from prior lines such as ToRL (Li et al., 2025b) or (Lin et al., 2025). The hierarchical decomposition is intuitively motivated but somewhat incremental relative to the existing literature, e.g., GiGPO (Feng et al., 2025) , and several details are under-specified.

**Strengths:**

• Clear Problem Motivation and Scope: The paper is grounded in well-known issues of reasoning—error propagation in long CoT traces and failures in symbolic computation—and formulates these in the RL framework (Sec. 2.1). The empirical correlation between code success and final correctness offers solid motivation for dual-level credit assignment.

• Hierarchical RL Formulation: The two-stage optimization (Eq. 4–9) represents a practical way to address sparse reward signals. Step-level RL on failed tool invocations is technically sensible and extends GRPO with fine-grained local feedback, offering a pragmatic, if not groundbreaking, advance.

**Weaknesses:**

• Data Generation Pipeline (TIRGen): The actor–critic construction (Alg. 1, Fig. 2) potentially yields a more noisy dataset than rule-based data due to the uncertainty introduced by LLMs. Althought multi-stage filtering avoids trivial tool use errors, I think human-in-the-loop verification and double-check is needed.

• Reproducibility and Transparency: Appendix D only offers implementation details, including batch sizes, optimizer settings, and prompt templates. But data and code are critical to support reproducibility and checking.

• Missing Analysis of Failure Modes: The paper does not report how often self-correction succeeds or fails, nor how many inference steps are affected by code errors or sandbox limitations (Sec. 2.4).

• Restricted Generalization Domain: All evaluations focus on mathematical or code generation. No evidence is given that THOR generalizes to broader tool environments (e.g., multi-turn search and agentic tasks).

**Questions:**

How to compute advantages at the step-level in Eq. 8? I think each sample contains a single think-act-observe loop, and the reward is computed from execution correctness. Do you mean the advantage is estimated by sampling a single response?

Moreover, does this method cause efficiency issue due the correctness evaluation and regeneration?

---

> ### Author Response · Authors · 2025-11-20
> **Author Response to Official Review by Reviewer RCoA (1/3)**
>
> We sincerely thank the reviewer RCoA for the constructive evaluation. Your recognition of our work is highly encouraging, and your comments have been invaluable in improving the clarity and technical depth of the manuscript.
>
> ### **1. W1: Data quality and human verification of TIRGen**
>
> > The actor–critic construction (Alg. 1, Fig. 2) potentially yields a more noisy dataset than rule-based data, the human-in-the-loop verification and double-check is needed.
>
> Thank you for the reviewer’s constructive comments. While an LLM-based actor–critic pipeline may theoretically introduce noise, our **multi-stage filtering mechanism** is specifically designed to ensure that only high-quality samples are retained.
>
> To further assess data quality beyond automated filtering, we **randomly sampled 100 examples** from the cold-start dataset and conducted a detailed human verification. The inspection evaluated:
>
> - (i) whether the transformation into tool-integrated reasoning steps was valid;
> - (ii) whether the generated code was both logically coherent and syntactically correct;
> - (iii) whether the execution outputs aligned with the reasoning process.
>
> The audit achieved a **100% pass rate**, providing strong evidence that the generated data maintains high fidelity. We additionally provide representative samples here: **[[Case of TIRGen](https://anonymous.4open.science/r/THOR_anonymous-161FX1/case_of_TIRGen_for_rebuttal.md)]**.
>
> Although adding human verification could further suppress noise, it would also impose substantial construction overhead and undermine scalability. One of our primary goals is to establish a **fully automated, policy-aligned data generation framework** that remains reusable across diverse models and experimental settings. In practice, the multi-stage filtering process—covering format validation, code-quality screening, and balancing for difficulty and tool-invocation depth—has proven highly effective at maintaining dataset quality **without requiring human intervention**.
>
> In summary, our human verification results and the multi-stage filtering mechanism jointly demonstrate that **TIRGen is able to generate high-quality training data while maintaining full automation and scalability**.
>
>
> ### **2. W2: reproducibility and transparency**
>
> > data and code are critical to support reproducibility and checking.
>
> Thank you for emphasizing the importance of reproducibility. We fully agree that transparent access to both data and code is crucial for validating and extending our contributions. To address this concern, we have prepared an **anonymous GitHub repository** that includes all implementation details and the full training dataset for verification purposes: **[[Our code and dataset](https://anonymous.4open.science/r/THOR_anonymous-161FX1)]**. The complete codebase and dataset will also be released publicly upon acceptance.
>
> We are currently refining and organizing the codebase, and will continue to progressively update the anonymous repository throughout the rebuttal period.
>
> ### **3. W3: More analysis of failure modes**
>
> > The paper does not report how often self-correction succeeds or fails, nor how many inference steps are affected by code errors or sandbox limitations.
>
> Thank you for the reviewer’s comments. We have added a quantitative analysis of failure modes with a focus on:
>
> - (i) the frequency of execution failures during tool-integrated reasoning,
> - (ii) the extent to which the self-correction mechanism mitigates these failures.
>
> For each benchmark, we computed the **number of failed code executions, successful executions, and the resulting code pass rate**, and compared the model’s behavior with and without the self-correction mechanism. The results are summarized below:
>
> | Method| Avg Acc (↑) | Code Ratio (↑) | Failed Code Num | Success Code Num | Code Pass Ratio (↑) |
> |-|-|-|-|-|-|
> | ***Non-Reasoning Models***| | ||||
> | Qwen2.5-Math-7B (Baseline)| 35.7% | 72.7% | 858 | 5806 | 87.1%|
> | THOR-7B | 59.7% | 96.6% | 685 | 5900 | 89.6%|
> | THOR-7B + self-correction | 61.2% | 96.2% | 82| 6050 | 98.7%|
> | ***Reasoning Models*** | | ||||
> | Qwen3-8B (Baseline) | 74.5% | 8.5%| 260 | 1071 | 80.5%|
> | THOR-Thinking-8B | 78.5% | 91.3% | 2311| 10535| 82.0%|
> | THOR-Thinking-8B + self-correction | 79.8% | 90.6% | 155 | 12532| 99.0%|
>
> From these results, we observe that:
>
> - The self-correction mechanism **substantially reduces** execution failures.
> - The **code pass rate rises sharply**, reaching 98–99% after self-correction. This demonstrates that our correction mechanism not only resolves code-level errors but also, to some extent, mitigates mistakes originating from faulty reasoning.
> - Task accuracy improves accordingly, reinforcing our central hypothesis that **successful code execution is strongly predictive of final answer correctness**.
>
> We have incorporated this analysis into Appendix C.2.2 (line 857-876).

---

> ### Author Response · Authors · 2025-11-20
> **Author Response to Official Review by Reviewer RCoA (2/3)**
>
> ### **4. W4: Generalization Beyond Mathematical and Code Tools**
>
> > No evidence is given that THOR generalizes to broader tool environments
>
> Thank you for the reviewer’s attention to the question of generalization. Although our empirical results focus on mathematical reasoning with a code-execution tool, we would like to emphasize that **the three core components of THOR are intrinsically tool-agnostic and are designed for a broad class of “interactive external tool environments.”** Concretely:
>
> 1. **Tool-agnostic design of TIRGen**
>    - TIRGen operates under a general pattern: the Actor proposes a reasoning step → the Critic determines whether an external tool can handle part of that step → the tool is executed and its feedback is used to refine the reasoning.
>    - This paradigm only assumes that the external tool provides **structured observations**, not that it is a code interpreter. By replacing the execution environment, TIRGen can be directly extended to search engines, retrieval APIs, and multi-step agent environments, requiring only minor adjustments to solvability detection and executor interfaces—**the overall framework remains unchanged**.
> 2. **Broad applicability of the hierarchical RL framework**
>    - Our hierarchical RL relies solely on (i) terminal rewards and (ii) intermediate feedback returned by the external tool.
>    - Tools such as web search, retrieval systems, or multi-turn APIs naturally provide verifiable intermediate signals (e.g., relevance scores, hit indicators, or structured API states). Hence, the optimization framework is **insensitive to the specific tool type** and ported to other settings without conceptual modification.
> 3. **Transferability of the self-correction mechanism**
>    - The self-correction mechanism hinges on local backtracking and regeneration triggered by external feedback.
>    - Any tool that provides explicit success/failure signals or interpretable intermediate feedback can similarly trigger self-correction, making the mechanism **inherently general across tool modalities**.
>
> In summary, our choice of mathematical tools stems from their controllability and verifiability, which enable rigorous analysis of THOR’s components. **The method itself remains structurally tool-agnostic and readily extensible to broader tool ecosystems (e.g., retrieval, search, multi-step API agents).** Extending THOR to additional tool environments is an important direction we plan to explore in future work.
>
> ### **5. Q1: Advantage computation**
>
> > How to compute advantages at the step-level in Eq. 8?
>
> Thank you for the reviewer’s question. Your understanding is correct. In step-level optimization, each sample contains **exactly one complete think–act–observe loop**, and its reward is directly determined by whether the executed code succeeds (0/1). The advantage computation at the step level follows the same normalized formulation as in the episode-level GRPO objective.
>
> Specifically, for a group of step-level rollouts sampled from the same prefix $\\{s_i'\\}_{i=1}^{G}$, the advantage is computed as:
> $$
> A_i^{'} = \frac{r_i - mean(r)}{std(r)}
> $$
> We have added a more detailed explanation of this advantage computation procedure in Section 2.3.2 (line 295-302) to prevent potential misunderstandings.

---

> ### Author Response · Authors · 2025-11-20
> **Author Response to Official Review by Reviewer RCoA (3/3)**
>
> ### **6. Q2: Efficiency analysis.**
>
> > does this method cause efficiency issue due the correctness evaluation and regeneration?
>
> Thank you for the reviewer’s question. In Appendix C.2.3 of the paper, we have already provided a token-level breakdown of inference cost. To further quantify the effect of correctness evaluation and regeneration, we additionally report **end-to-end inference time (seconds) and FPS** across all 6,184 samples. All experiments are performed using the vLLM inference engine on a single machine with 8×H200 GPUs.
>
> | Method| MATH 500 | AIME 2024 | AIME 2025 | AMC 2023 | Minerva Math | Olympiad Bench | Avg Tokens | Total Infer Seconds (↓) | FPS (↑)|
> | - | - | - | - | - | - | - | - | - | - |
> | ***Non-Reasoning Models*** ||||||  ||| |
> | Qwen2.5-Math-7B| 866 | 1283 | 1325 | 1132| 802| 1090| 1083| 1379 | 4.48 |
> | THOR-7B | 705 | 1351 | 1420 | 928 | 729| 981| 1019| 1324 | 4.67 (↑4%)  |
> | ***Reasoning Models***||||||  ||| |
> | Qwen3-8B| 5102| 11986| 13022| 7989| 6906  | 9238| 9041| 7244 | 0.85 |
> | THOR-Thinking-8B| 4506| 10338| 11807| 6749| 5444  | 8205| 7841| 6280 | 0.98 (↑15%) |
>
> From these results, we observe that:
>
> - **Self-correction introduces virtually no additional inference overhead**.
> - **THOR yields improved inference efficiency**: with tool-integrated guidance, the model produces substantially shorter reasoning chains, resulting in lower total inference cost.
>
> Thus, although self-correction involves correctness evaluation and regeneration, the overall runtime impact is negligible. In practice, THOR **reduces** inference cost due to more concise reasoning.
>
> These additional analyses have been incorporated into Appendix C.2.3 (line 878-899) of the main paper.

---

> ### Author Response · Authors · 2025-11-25
> **Looking Forward to Your Response**
>
> Dear reviewer RCoA,
>
> Thank you again for the time you have devoted to reviewing our submission and our rebuttal. As the discussion period is drawing to a close, we would greatly appreciate it if you could let us know whether our responses have addressed your concerns, or if any further clarification would be helpful on our side.
>
>
> Best regards,
>
> Authors of submission 5050

---

> > ### Comment · Reviewer_RCoA · 2025-11-28
> > **My concerns have been addressed**
> >
> > Thanks for the detailed response. And I tend to accept this work.

---

> > > ### Author Response · Authors · 2025-11-28
> > > **Thank you for your follow-up**
> > >
> > > Dear reviewer RCoA,
> > >
> > > Thank you very much for your **positive follow-up** and for taking the time to carefully review our rebuttal. We are glad to hear that **our clarifications have addressed your concerns**. If there is anything else we can further elaborate on during the remaining discussion period, we would be happy to provide additional details.
> > >
> > > We truly appreciate your constructive feedback and consideration.
> > >
> > > Best regards,
> > >
> > > Authors of submission 5050

---

### Official Review · Reviewer_5muf · 2025-10-31

**Soundness:** 2
**Presentation:** 3
**Contribution:** 2
**Rating:** 6
**Confidence:** 4

**Summary:**

This paper tackles the poor computational and symbolic reasoning of LLMs in math problems. It introduces THOR, a framework built on three key parts. 1) TIRGen: A data pipeline described as a multi-agent "actor-critic" system, for generating high-quality examples of tool-integrated reasoning. 2) Hierarchical RL: A two-level optimization strategy. It combines a sparse, "episode-level" reward for the final answer's correctness with a denser, "step-level" reward for the success of intermediate code generation. 3) Self-Correction: An inference-time mechanism that lets the model backtrack and revise its reasoning path when a tool call fails.

**Strengths:**

1. Novel and Effective RL Framework: The paper's core strength lies in its hierarchical reinforcement learning strategy, which decisively tackles the sparse-reward problem in complex reasoning by introducing a denser, intermediate reward signal (code execution success), all while maintaining the focus on the primary objective (final answer correctness). This approach is both elegant and purposeful.
2.  Strong Empirical Performance: The paper achieves state-of-the-art results on several challenging math benchmarks (Tab 1). These gains, particularly against the non-reasoning model baseline, underscore the system's high efficacy.
3. Comprehensive Ablation Study: The ablation analysis in Table 3 systematically exposes each component's significant and additive impact—TIRGen dataset (T3 vs T1), hierarchical RL (T5 vs T3), and self-correction (T6 vs T5). This compellingly supports the paper's claims.

**Weaknesses:**

1.  Could the authors please clarify the choice of the term "Actor-Critic" for TIRGen? Since the "Critic" model converts reasoning steps into code, I am curious whether there are reasons to favor this RL-inspired terminology over alternatives such as "Generator-Refiner" or "Proposer-Converter."

2.   In the context of model evaluation, the Appendix indicates that the "Critic" is a 32B model, whereas the "Actor" is an 8B or 1.5B model. Could the authors comment on the importance of this size difference? Additionally, how might the quality of the TIRGen dataset ($\mathcal{D}_{SFT}$) be affected if the Critic model were the same size as (or smaller than) the Actor model?

3.  The self-correction mechanism (Section 2.4) is triggered when code execution fails. I am interested in how the framework deals with logically incorrect code that still executes successfully (e.g., producing a number from the wrong equation). Would an episode-level RL objective suffice to address this, or is an additional mechanism required?

**Questions:**

See weakness.

---

> ### Author Response · Authors · 2025-11-20
> **Author Response to Official Review by Reviewer 5muf (1/2)**
>
> Thank you very much to reviewer 5muf for your attention to and recognition of our work. Your feedback is greatly encouraging to us. We appreciate the clarity and constructiveness of your comments, which help us further improve the paper.
>
> ### **1. W1: the choice of the term "Actor-Critic" for TIRGen.**
>
> > clarify the choice of the term "Actor-Critic" for TIRGen instead of "Generator-Refiner" or "Proposer-Converter".
>
> Thank you very much for your valuable feedback. We fully agree that the term **"Generator-Refiner"** is more precise and intuitive than "Actor-Critic" for describing our data construction pipeline.
>
> Our original use of "Actor-Critic" was metaphorically inspired by the iterative interaction loop in RL. However, as the reviewer correctly pointed out, our “Critic” agent specifically focuses on **identifying codable steps and converting them into executable code**, rather than estimating a value function (scalar reward) as in standard RL literature.
>
> To improve clarity and avoid confusion with the actual Reinforcement Learning training phase (Section 2.3), we now represent the functional roles more accurately by renaming the components. In the revised manuscript, we have updated all mentions of “Actor” to **“Generator”** and “Critic” to **“Refiner”** within the paper (line 17-19, 81-86, 145-147, 156-193, 504-506, 910-912, 961-963, 1079, 1086).
>
> For consistency with the reviewer’s question, we temporarily retain the term “Actor-Critic” in the Comments below.
>
> ### **2. W2: Ablation on different sizes of Actor and Critic models.**
>
> > how might the quality of the TIRGen dataset ($D_{SFT}$) be affected if the Critic model were the same size as (or smaller than) the Actor model ?
>
> Thank you very much for your interest in our work. We further analyze how different Critic models affect data quality.
>
> To isolate the effect of the Critic itself, we re-run the original TIRGen on standard benchmarks **without any multi-stage filtering**, allowing us to directly observe how different Critic sizes influence the raw generated data. The results are summarized below:
>
> | Actor-Model| Critic-Model | MATH 500 | AIME 2024 | AIME 2025 | AMC 2023 | Minerva Math | Olympiad Bench | Avg Acc| Code Ratio|
> |-|-|-|-|-|-|-|-|-|-|
> | ***Non-Reasoning Models*** |||||||| | |
> | Qwen2.5-7B-Math| Qwen3-1.7B| 67.1| 14.2 | 12.5 | 51.3| 30.6| 40.3| 36.0 | 60.3 |
> | Qwen2.5-7B-Math| Qwen3-4B| 72.7| 22.5 | 12.5 | 53.1| 36.5| 44.8| 40.4 | 79.2 |
> | Qwen2.5-7B-Math| Qwen3-8B| 73.8| 25.8 | 13.3 | 60.6| 38.1| 45.1| **42.8**| **88.8**|
> | Qwen2.5-7B-Math| Qwen3-32B| 73.0| 23.3 | 12.5 | 61.3| 37.0| 45.5| 42.1 | 85.8 |
> | ***Reasoning Models***|||||||| | |
> | Qwen3-8B| Qwen3-1.7B| 88.0| 35.7 | 25.0 | 71.5| 55.6| 63.6| 56.6 | 42.7 |
> | Qwen3-8B| Qwen3-4B| 89.3| 40.8 | 30.8 | 76.9| 57.8| 66.8| 60.4 | 41.2 |
> | Qwen3-8B| Qwen3-8B| 90.0| 50.8 | 31.1 | 72.5| 57.7| 66.6| 61.5 | 46.4 |
> | Qwen3-8B| Qwen3-32B| 91.7| 52.5 | 40.0 | 85.0| 60.0| 71.6| **66.8**| **52.5**|
>
> From the experimental results, we observe:
>
> 1. **Non-reasoning models benefit from similar-sized Critics**
>
>    - For non-reasoning models (e.g., Qwen2.5-Math-7B), the Actor exhibits short-CoT reasoning, simple local logic, and clearly defined codable structures. In this setting, the primary objective of TIRGen is to maintain **policy alignment**, meaning that the code-augmented steps produced by the Critic must strictly adhere to the Actor’s inherent linguistic style and reasoning pattern.
>
>    - When the Critic is too strong (e.g., Qwen3-32B), **it tends to rewrite or over-strengthen reasoning or coding abilities that the Actor does not inherently possess, thereby reducing the quality of the generated dataset**. A model around the Qwen3-8B scale is already sufficient for stable and accurate code conversion.
>
> 2. **Reasoning models require stronger Critics to handle long, complex chains**
>
>    - For reasoning models (e.g., Qwen3-8B-Thinking), the Actor produces long CoT sequences with deep logical structure and multiple codable components. In this case, **the Critic must reliably extract the logical core, identify executable segments, and generate high-quality code**.
>
>    - The complexity of this extraction task increases substantially, and larger Critics show clear performance advantages.
>
> 3. This phenomenon also indicates that **our method does not distill mathematical ability from a larger model, but relies solely on its robustness in instruction following and code generation**.
>
> Importantly, there is **no need to worry about final performance divergence across Critics**. We have designed a highly refined multi-stage data filtering pipeline that ensures the effect of using different Critics (Qwen3-8B vs. Qwen3-32B) is minimized after cold start. The code for data construction and filtering will also be released (in response to Reviewer RCoA: **2.W2: reproducibility and transparency**).

---

> ### Author Response · Authors · 2025-11-20
> **Author Response to Official Review by Reviewer 5muf (2/2)**
>
> ### **3. W3: how the framework deals with logically incorrect code that still executes successfully.**
>
> > How the framework deals with logically incorrect code that still executes successfully? Would an episode-level RL objective suffice to address this, or is an additional mechanism required?
>
> Thank you for the reviewer’s attention to our work.
>
> 1. **Episode-level RL can partially mitigate such errors**
>
>    Since the final reward is determined solely by the correctness of the final answer, logically incorrect but executable code typically still leads to an incorrect answer and thus receives negative feedback at the trajectory level. Therefore, episode-level RL provides a certain degree of suppression for this type of error.
>
> 2. **To achieve a more effective step-level supervision method, we are exploring a new fine-grained reward shaping mechanism**
>
>    To further improve performance, we are currently **exploring a fine-grained reward shaping method for evaluating the contribution of each code block to the final answer**. The high-level idea is as follows:
>
>    - Truncate the reasoning chain at various positions and insert a lightweight probe (e.g., a “final answer \boxed{” query) to assess the model’s predicted final answer from that intermediate state.
>
>    - Track how the predicted answer shifts as the reasoning progresses to estimate each step’s marginal contribution.
>
>    - In TIR, **by comparing the model’s predicted-answer states before and after each code call, we can evaluate the correctness, necessity, and timing of the tool invocation, compensating for the lack of feedback when execution succeeds**.
>
>    Early experiments under a pure CoT setting (R1-Distill-Qwen2.5-7B) already show encouraging improvements. In future work, we plan to integrate this reward shaping approach into the THOR training framework for further improvement.

---

> ### Author Response · Authors · 2025-11-25
> **Looking Forward to Your Response**
>
> Dear reviewer 5muf,
>
> Thank you again for the time you have devoted to reviewing our submission and our rebuttal. As the discussion period is drawing to a close, we would greatly appreciate it if you could let us know whether our responses have addressed your concerns, or if any further clarification would be helpful on our side.
>
>
> Best regards,
>
> Authors of submission 5050

---

### Official Review · Reviewer_ybnx · 2025-10-31

**Soundness:** 2
**Presentation:** 2
**Contribution:** 2
**Rating:** 4
**Confidence:** 4

**Summary:**

This paper proposes a tool-integrated reasoning framework with three components: (i) a multi-agent pipeline that builds tool-integrated reasoning (TIR) trajectories for cold-start SFT; (ii) hierarchical RL that jointly optimizes episode-level answer accuracy and step-level code-pass rewards; and (iii) an inference-time self-correction mechanism that backtracks to the last failed tool-use step.

**Strengths:**

1. The writing is clear.
2. The method shows gains over recent TIR/CoT baselines.

**Weaknesses:**

1. The method assumes a failed execution implies both the action and its preceding reasoning step are wrong, and always backtracks/regenerates the suffix. This coupling can be brittle when failures arise from syntax/runtime issues; action-only repair may suffice.
2. Optimizing only code-pass is not a promising step for fine-grained TIR reward shaping; analyses of code-trigger timing, necessity/helpfulness, and over/under-calling would be more informative.
3. The self-correction trigger is hand-coded, whereas prior work such as ReTool [2] reports emergent self-correction behavior during training.
4. TIR data curation and RL for tool use have seen substantial prior exploration[1,2,3], while the experiments and design here are solid, the technical novelty appears incremental relative to recent work.

[1] ToRL: Scaling Tool-Integrated RL

[2] ReTool: Reinforcement Learning for Strategic Tool Use in LLMs

[3] ToolRL: Reward is All Tool Learning Needs

**Questions:**

1. How often do failures stem from action errors versus reasoning errors? Please report an ablation: (a) action-only repair, (b) step-suffix repair, (c) full re-plan.
2. In Table 2, how does your self-rewarded test-time selection compare with BoN/self-consistency and PRM/verifier-based reranking under the same (N), temperature, and compute?
3. Does optimizing code-pass reward improve code-generation ability itself, or only downstream answer accuracy?
4. Is the training data comparable to baselines (tokens, steps, RL updates)?
5. Could you add more TIR baselines on the same backbone and training setup to isolate method effects?

---

> ### Author Response · Authors · 2025-11-20
> **Author Response to Official Review by Reviewer ybnx (1/5)**
>
> We thank the reviewer ybnx for the clear summary and constructive suggestions. We appreciate the strengths you highlighted and have conducted additional analyses to directly address the concerns regarding failure attribution, fine-grained rewards, self-correction triggers, and baseline comparisons.
>
> ### **1. W1 & Q1: Analysis of Execution Failures and Repair Strategies Ablation**
>
> > The self-correction can be brittle when failures arise from syntax/runtime issues; action-only repair may suffice. How often do failures stem from action errors versus reasoning errors? Please report an ablation: (a) action-only repair, (b) step-suffix repair, (c) full re-plan.
>
> Thank you for raising this important question. We fully agree with your observation.
>
> Execution failures do not always imply that the entire reasoning chain is incorrect; some failures indeed arise purely from syntax or runtime errors in the generated code, for which action-only repair may suffice. However, beyond these surface-level code issues, **a substantial portion of failures originate from upstream logical reasoning errors**, which motivates our design of suffix repair to more robustly address diverse failure modes.
>
> To systematically assess the effectiveness of different repair strategies, we additionally evaluate three variants: (a) action-only repair, (b) step-suffix repair, and (c) full re-plan.
>
> From the results, we find that **step-suffix repair consistently outperforms action-only repair**, indicating that our mechanism effectively handles both code-level errors and deeper reasoning-level inconsistencies. In contrast, full re-plan tends to overwrite correct earlier reasoning when the failure is merely local (e.g., a minor syntax error), which can lead to unnecessary perturbation and degraded performance.
>
> These additional experimental results have been incorporated into Appendix C.2.1 (line 836-855) of the paper.
>
> | Model | MATH 500 | AIME 2024 | AIME 2025 | AMC 2023 | Minerva Math | Olympiad Bench | Avg |
> | - | - | - | - | - | - | - | - |
> | ***Non-Reasoning Models*** || | |||| |
> | THOR-7B | 87.3 | 45.0 | 31.7 | 80.0 | 53.4 | 60.5 | 59.7 |
> | THOR-7B + action-only repair | 87.0 | 46.3 | 32.1 | 82.5 | 53.3 | 60.6 | 60.3 |
> | THOR-7B + step-suffix repair (Ours) | 87.5 | 50.0 | 33.3 | 81.3 | 53.9 | 61.1 | **61.2** |
> | THOR-7B + full re-plan | 86.6| 46.7 | 34.2 | 81.3| 52.2  | 61.1| 60.4 |
> | ***Reasoning Models*** ||||||  ||
> | THOR-Thinking-8B | 96.6 | 74.2 | 60.0 | 95.6 | 65.4 | 79.0 | 78.5 |
> | THOR-Thinking-8B + action-only repair | 96.8 | 75.0 | 60.0 | 95.6 | 64.0 | 79.7 | 78.5 |
> | THOR-Thinking-8B + step-suffix repair (Ours) | 96.8 | 77.5 | 62.5 | 96.8 | 65.6 | 79.7 | **79.8** |
> | THOR-Thinking-8B + full re-plan | 96.2| 75.8 | 62.5 | 98.1| 63.4  | 79.3| 79.2 |

---

> ### Author Response · Authors · 2025-11-20
> **Author Response to Official Review by Reviewer ybnx (2/5)**
>
> ### **2. W2: Clarifying the Design Choice of Code-Pass Reward**
>
> > Optimizing only code-pass is not a promising step; analyses of code-trigger timing, necessity/helpfulness, and over/under-calling would be more informative.
>
> We fully agree that more fine-grained analysis of tool invocation is valuable. However, this work focuses on a **fine-grained yet supervision-free optimization design** that already yields substantial gains. Below, we clarify why we adopt code-pass as the reward signal and outline our ongoing exploration of richer reward shaping strategies.
>
> 1. **Code-pass is currently the simplest and most effective fine-grained reward signal**
>
>     - Its strong correlation with final accuracy is statistically validated (Appendix C.1).
>
>     - It requires no external critic/PRM or symbolic verifier, adds no extra cost, and scales stably, yielding clear improvements (Table 2 & 3).
>
> 2. **We did not model “code-trigger timing” or “necessity/helpfulness” due to scope and cost, not because they are unimportant**
>
>    We agree these factors matter, but they introduce significant practical challenges:
>
>     - These metrics require additional Critic/PRM evaluation, adding instability and inference overhead.
>
>     - Detecting over-/under-calling behavior demands large-scale trajectory statistics, substantially expanding the scope of the work.
>
> 3. **This direction is highly promising and is precisely one of our current research focuses**
>
>    Your suggestion is both reasonable and important. To further improve performance, we are currently exploring a fine-grained reward shaping method for evaluating the contribution of each code block to the final answer. The high-level idea is as follows:
>
>     - Truncate the reasoning chain at various positions and insert a lightweight probe (e.g., a “final answer \boxed{” query) to assess the model’s predicted final answer from that intermediate state.
>
>     - Track how the predicted answer shifts as the reasoning progresses to estimate each step’s marginal contribution.
>
>     - In TIR, **by comparing the model’s predicted-answer states before and after each code call, we can evaluate the correctness, necessity, and timing of the tool invocation, compensating for the lack of feedback when execution succeeds**.
>
>    Early experiments under a pure CoT setting (R1-Distill-Qwen2.5-7B) already show encouraging improvements. We plan to integrate this reward shaping approach into the THOR training framework in the future.
>
> Summary: We view these additional reward signals as complementary to the approach proposed in this paper. Our primary contribution is to show that **even a simple signal such as code-pass is sufficient to achieve stable TIR, providing a solid foundation upon which richer reward shaping can be further developed.**
>
> ### **3. W3: On the Necessity of Explicit Self-Correction in TIR**
>
> > prior work such as ReTool reports emergent self-correction behavior during training.
>
> We appreciate the reviewer’s reference to ReTool, which is indeed **an insightful and well-executed study** demonstrating that strategic tool use can emerge through reinforcement learning. However, our empirical findings show that **in TIR settings, emergent self-correction alone is unreliable, whereas an explicit triggering mechanism is both necessary and effective**.
>
> Although prior work (e.g., ReTool, Qwen2.5-Instruct, Qwen3) reports emergent self-correction in pure CoT reasoning, we find that this behavior **does not reliably transfer** to TIR. Under consecutive execution failures, models frequently, models often **abandon tool use and fall back to CoT-only reasoning** **[[Case Study Here](https://anonymous.4open.science/r/THOR_anonymous-161FX1/case_for_rebuttal.md)]**, showing that emergent correction alone cannot ensure stable tool integration.
>
> assess the benefit of explicit correction, we compare the Baseline, THOR-7B (emergent correction), and THOR-7B with our explicit mechanism across **accuracy, code ratio, failed/successful code blocks, and code pass rate**. Explicit correction **substantially reduces failures**, **significantly boosts code pass rate**, and consistently improves accuracy. Notably, **the instability of emergent correction holds across multiple TIR baselines**, highlighting the practical need for an explicit mechanism.
>
> These additional analyses have been incorporated into Appendix C.2.2 (line 857-876).
>
> |Method|Avg Acc (↑)|Code Ratio (↑)|Failed Code Num|Success Code Num|Code Pass Ratio (↑)|
> |-|-|-|-|-|-|
> |***Non-Reasoning Models***||||||
> |Qwen2.5-Math-7B (Baseline)|35.7%| 72.7%|858| 5806 |87.1%|
> |THOR-7B (emergent self-correction)|59.7%|96.6%|685|5900|89.6%|
> |THOR-7B + Our hand-coded self-correction|61.2%| 96.2%|82|6050|98.7%|
> |***Reasoning Models*** ||||||
> |Qwen3-8B (Baseline)|74.5%|8.5%|260|1071|80.5%|
> |THOR-Thinking-8B (emergent self-correction)|78.5%| 91.3%|2311|10535|82.0%|
> |THOR-Thinking-8B + Our hand-coded self-correction|79.8%|90.6%|155|12532|99.0%|

---

> ### Author Response · Authors · 2025-11-20
> **Author Response to Official Review by Reviewer ybnx (3/5)**
>
> ### **4. W4: Clarifying the Novelty of THOR Beyond Prior TIR Data Curation and RL Work**
>
> >the technical novelty appears incremental relative to recent work.
>
> Thank you for the reviewer’s comments. Although THOR builds on prior work in TIR data construction and tool-integrated RL, our contributions **tackle two unresolved core bottlenecks**.
>
> 1. **Existing TIR data synthesis methods fail on reasoning models and even shrink their capability frontier**
>
>    - Prior TIR data construction approaches [1,2,3] implicitly **assume that models already possess stable tool-use behavior**, and thus are **mainly effective only for non-reasoning models**.
>    - They break down on reasoning models and cannot reliably induce TIR behaviors.
>
>    To overcome this limitation, we introduce TIRGen, a fully automated TIR data construction pipeline that, for the first time, provides:
>
>    - **A unified data synthesis pipeline applicable to both reasoning and non-reasoning models**
>
>    The table below compares the cold-start performance of different data-construction methods. TIRGen successfully enables both reasoning and non-reasoning models to acquire TIR abilities, substantially expanding their capability frontier.
>
>    | Method| Pass@16| Code Ratio|
>    |-|-|-|
>    | ***Non-Reasoning Models*** | | |
>    | Qwen2.5-Math-7B| 72.0 | 74.8 |
>    | Qwen2.5-Math-7B + TIRGen (Ours) | **79.7**| **91.7**|
>    | Qwen2.5-Math-7B + ReTool [2]| 66.4 | 43.6 |
>    | ***Reasoning Models***| | |
>    | Qwen3-8B| 87.6 | 8.5|
>    | Qwen3-8B + TIRGen (Ours) | **90.9**| **88.3**|
>    | Qwen3-8B + Nemotron [4]| 84.1 | 81.3 |
>
> 2. **We are the first to use code-pass as a unified fine-grained signal for both RL training and inference**
>
>    Prior works [1,2,3] rely exclusively on episode-level rewards based on the final answer, **discarding the rich feedback available from intermediate tool calls**. In contrast, THOR introduces:
>
>    - A unified fine-grained signal, code-pass.
>
>    - A hierarchical RL framework that combines episode-level and step-level rewards.
>
>    - Online self-correction during inference.
>
>    Under fair and comparable settings with models of the same scale, THOR achieves an average improvement of 5–10% over ToRL, ZTRL, and AutoTIR, covering both reasoning and non-reasoning models.
>
> [1] ToRL: Scaling Tool-Integrated RL
>
> [2] ReTool: Reinforcement Learning for Strategic Tool Use in LLMs
>
> [3] ToolRL: Reward is All Tool Learning Needs
>
> [4] AIMO-2 Winning Solution: Building State-of-the-Art Mathematical Reasoning Models with OpenMathReasoning dataset
>
> ### **5. Q1: ablation on different repair methods.**
>
> Same as **1. W1 & Q1: Analysis of Execution Failures and Repair Strategies Ablation**.
>
> ### **6. Q2: self-rewarded test-time selection compared with BoN/self-consistency and PRM/verifier-based reranking**
>
> > self-rewarded test-time selection compare with BoN/self-consistency and PRM/verifier-based reranking.
>
> We have added a comprehensive comparison under **the same sampling budget (N = 8), identical temperature settings, and equal inference compute**, including self-consistency [5], InternLM2-RM [6], Qwen2.5-Math-PRM [7], and our self-rewarded test-time selection (Section 3.3).
>
> We also evaluate a **hybrid strategy that combines our self-rewarded selection with PRMs**. When N is large, multiple samples may share the same code-pass rate; in such cases, PRMs can provide an additional preference signal to further differentiate among candidates and select better reasoning paths.
>
> | Model| MATH 500 | AIME 2024 | AIME 2025 | AMC 2023 | Minerva Math | Olympiad Bench | Avg |
> |-|-|-|-|-|-|-|-|
> | ***Non-Reasoning Models***||||||||
> | THOR-7B| 87.5| 50.0 | 33.3 | 81.3| 53.9| 61.1| 61.2|
> | THOR-7B + Self-Consistency [5] | 89.8| 50.0 | 36.7 | 80.0| 56.3| 64.8| 62.9|
> | THOR-7B + InternLM2-RM [6]| 88.2| 56.7 | 36.7 | 77.5| 55.9| 62.8| 63.0|
> | THOR-7B + Qwen2.5-Math-PRM [7] | 89.6| 53.3 | 33.3 | 77.5| 57.7| 66.5| 63.0|
> | THOR-7B + self-rewarded (**Ours**) | 87.7| 53.3 | 38.3 | 83.8| 54.9| 61.5| 63.3|
> | THOR-7B + InternLM2-RM [6] + self-rewarded (**Ours**)| 88.0| 60.0 | 33.3 | 85.0| 56.3| 63.5| **64.4** |
> | ***Reasoning Models***||||||||
> | THOR-Thinking-8B | 96.8| 77.5 | 62.5 | 96.8| 65.6| 79.7| 79.8|
> | THOR-Thinking-8B + Self-Consistency [5] | 97.4| 90.0 | 66.7 | 95.0| 63.6| 81.6| 82.4|
> | THOR-Thinking-8B + InternLM2-RM [6]| 96.8| 83.3 | 73.3 | 97.5| 66.5| 81.5| 83.2|
> | THOR-Thinking-8B + Qwen2.5-Math-PRM [7] | 97.0| 73.3 | 63.3 | 95.0| 64.7| 80.4| 79.0|
> | THOR-Thinking-8B+ self-rewarded (**Ours**)| 97.2| 86.7 | 70.0 | 97.5| 65.8| 82.0| 83.2|
> | THOR-Thinking-8B + InternLM2-RM [6] + self-rewarded (**Ours**) | 98.0| 86.7 | 73.3 | 100.0| 65.4| 82.1| **84.3** |
>
> See the rebuttal for Q2 in the next response. ↓

---

> ### Author Response · Authors · 2025-11-20
> **Author Response to Official Review by Reviewer ybnx (4/5)**
>
> We continue the rebuttal for Q2:
>
> From the experimental results, we observe:
>
> - Our self-rewarded Inference Enhancement **already achieves performance comparable to or even better than strong PRMs without requiring any external PRM/verifier**.
> - **When combined with PRMs, the self-rewarded method achieves further improvements**, indicating that it provides a supervision signal complementary to PRMs.
> - **Our reward signal is not sensitive to the model type**: it consistently improves both reasoning and non-reasoning models. In contrast, PRMs often suffer from limited training coverage, leading to unstable transfer.
>
> Overall, the proposed self-rewarded inference enhancement offers a simple, lightweight, and scalable test-time strategy that remains effective across diverse reasoning paradigms.
>
> We have included these results in Section 3.3 (Table 2) (line 393-422).
>
> [5] Universal Self-Consistency for Large Language Model Generation
>
> [6] InternLM2 Technical Report
>
> [7] The Lessons of Developing Process Reward Models in Mathematical Reasoning
>
> ### **7. Q3: Generalization on code-generation ability.**
>
> > Does optimizing code-pass reward improve code-generation ability itself ?
>
> Thank you for the reviewer’s question. While the main paper (Figure 5) already reports how hierarchical RL strengthens code generation, here we provide a more complete set of evaluations.
>
> We assess THOR on three widely used code benchmarks—HumanEval+, MBPP+, and LiveCodeBench—to quantify its impact on code generation ability. The results are consistent: **for both reasoning and non-reasoning models, optimizing the code-pass reward not only improves mathematical answer accuracy but also substantially enhances intrinsic code generation ability**.
>
> | Model | Human Eval+ | MBPP+ | Live Code Bench | Avg|
> | - | - | - | - | - |
> | ***Non-Reasoning Models*** | ||| |
> | Qwen2.5-Math-7B| 53.7 | 57.4| 12.0| 40.1 |
> | THOR-7B (Ours) | 59.8 | 60.6| 19.4| 45.0 |
> | ***Reasoning Models***| ||| |
> | Qwen3-8B| 83.5 | 68.8| 49.1| 67.3 |
> | THOR-Thinking-8B (Ours)| 84.8 | 71.2| 52.6| 69.5 |
>
> ### **8. Q4: comparable of training data with other baselines.**
>
> > Is the training data comparable to baselines ?
>
> Yes. The overall training scale of THOR is **fully comparable to major baselines** in terms of SFT data size, RL query count, and RL update steps. To ensure fairness, we surveyed the training configurations of both CoT-based and TIR-based methods and aligned them with our setup. A summary is provided in the table below.
>
> | Method| SFT Sample Num | RL Query Num | RL steps |
> | - | - | - | - |
> | ***Non-Reasoning Models*** ||||
> | Eurus-2-PRIME-7B| 229,763 | 480,537 | 592 |
> | TORL-7B | -| 28,740| 1000|
> | AutoTIR-7B| -| 46,657| 250 |
> | ZTRL-7B | -| 159,900 | 600 |
> | ARTIST-7B [8]| -| 20,000| 1000|
> | TATA-7B [9]| 483,000 | -| - |
> | THOR-7B (Ours) | 29,217| 28,740| 800 |
> | ***Reasoning Models***||||
> | DeepScaleR-1.5B| -| 40,315| 1750|
> | START-32B| 49,969| -| - |
> | OpenMath-Nemotron-7B| 5,466,000 | -| - |
> | THOR-Thinking-8B| 57,598| 28,740| 800 |
>
> From the comparison, we observe:
>
> - **SFT data size**: THOR uses no more data than typical baselines, and in fact **less than some methods** (e.g., TATA, OpenMath-Nemotron).
> - **RL query count**: THOR uses **exactly the same number of RL queries as ToRL** (28,740).
> - **RL update steps**: THOR’s update steps **fall squarely within the common range** (250–1750).
>
> Therefore, the training scale of THOR is **fair and comparable** to other baselines.
>
> [8] Agentic Reasoning and Tool Integration for LLMs via RL
>
> [9] Teaching llms according to their aptitude: Adaptive reasoning for mathematical problem solving

---

> ### Author Response · Authors · 2025-11-20
> **Author Response to Official Review by Reviewer ybnx (5/5)**
>
> ### **9. Q5: add more TIR baselines on the same backbone and training setup.**
>
> > add more TIR baselines on the same backbone and training setup.
>
> Thank you for the reviewer’s suggestion. We have added more TIR baselines, all of which use the same or highly similar backbones (Qwen2.5-Math-7B / Qwen2.5-7B series) and comparable training paradigms (SFT/RFT/RL). We unified the training configurations, backbone consistency, and performance across mathematical benchmarks, with the results summarized in the table below:
>
> | Method| Training setup | Backbone| MATH 500 | AIME 2024 | AIME 2025 | AMC 2023 | Minerva Math | Olympiad Bench | Avg|
> | - | - | - | - | - | - | - | - | - | - |
> | ***Non-Reasoning Models*** ||||||||| |
> | TORL-7B | RL | Qwen2.5-Math-7B| 82.2| 40.8 | 29.2 | 73.8| 53.8| 57.3| 56.2 |
> | AutoTIR-7B| RL | Qwen2.5-7B-Instruct | 62.6| 33.3 | 16.7 | - | -| -| -|
> | ZTRL-7B | RL | Qwen2.5-7B| 80.2| 50.0 | 26.7 | - | -| -| -|
> | ARTIST-7B [8]| RL | Qwen2.5-7B-Instruct | 67.6| 15.6 | -| 47.0| -| 37.9| -|
> | TATA-7B [9]| SFT + RFT | Qwen2.5-Math-7B| 73.0| -| -| - | -| 35.9| -|
> | THOR-7B (Ours) | SFT + RL| Qwen2.5-Math-7B| 87.5| 50.0 | 33.3 | 81.3| 53.9| 61.1| 61.2 |
> | ***Reasoning Models***||||||||| |
> | START-32B| RFT| QwQ-32B-Preview| 94.4| 66.7 | 47.1 | 95.0| -| -| -|
> | OpenMath-Nemotron-7B| SFT + RFT | Qwen2.5-Math-7B| - | 72.9 | 57.5 | - | -| -| -|
> | THOR-Thinking-8B| SFT + RL| Qwen3-8B| 96.8| 77.5 | 62.5 | 96.8| 65.6| 79.7| 79.8 |
>
> From the results, we observe:
>
> - Under the same backbone (Qwen2.5-Math-7B), **THOR-7B consistently outperforms all comparable TIR baselines** (TORL, ZTRL, AutoTIR, TATA, ARTIST).
> - For reasoning models, **THOR-Thinking-8B also significantly outperforms OpenMath-Nemotron and START under the same or even smaller training cost**.
>
> These additional comparisons have been included in Table 1 of the paper (line 336-338, 371-374).
>
> [8] Agentic Reasoning and Tool Integration for LLMs via RL
>
> [9] Teaching llms according to their aptitude: Adaptive reasoning for mathematical problem solving

---

> ### Author Response · Authors · 2025-11-25
> **Looking Forward to Your Response**
>
> Dear reviewer ybnx,
>
> Thank you again for the time you have devoted to reviewing our submission and our rebuttal. As the discussion period is drawing to a close, we would greatly appreciate it if you could let us know whether our responses have addressed your concerns, or if any further clarification would be helpful on our side.
>
>
> Best regards,
>
> Authors of submission 5050

---

> ### Author Response · Authors · 2025-11-28
> **Kindly Following Up on Your Feedback**
>
> Dear reviewer ybnx,
>
> I hope this message finds you well. With **less than a week** left in the discussion period, we would like to kindly follow up on whether our rebuttal has addressed your concerns. If any clarification is still needed, we would be glad to provide it.
>
> Thank you again for the time and effort. We sincerely look forward to your response.
>
> Best regards,
>
> Authors of submission 5050

---

### Author Response · Authors · 2025-11-30
**Summary of Paper Revision**

In response to the reviewers’ comments, we have revised the manuscript accordingly, with all modifications highlighted in blue. The major changes are as follows:

**Main Paper:**
- **Section 2.3.2 (line 295-302)**: Added a more detailed explanation of the advantage computation procedure. (Reviewer RCoA Q1)
- **Section 3.3 Table 2 (line 393-422)**: Added comparisons between our self-rewarded test-time selection and BoN/self-consistency as well as PRM/verifier-based reranking methods. (Reviewer ybnx Q2)
- **Section 3.2 Table 1 (line 336-338, 371-374)**: Added additional TIR baselines using the same backbone and training configuration. (Reviewer ybnx Q5)
- **Throughout the paper (line 17-19, 81-86, 145-147, 156-193, 504-506, 910-912, 961-963, 1079, 1086)**: Updated all terminology, replacing “Actor” with “Generator” and “Critic” with “Refiner” for improved clarity. (Reviewer 5muf W1)

**Appendix:**
- **Appendix C.2.1 (line 836-855)**: Added an ablation study comparing different repair strategies. (Reviewer ybnx W1&Q1)
- **Appendix C.2.2 (line 857-876)**: Added analysis comparing the Baseline and THOR-7B across accuracy, code ratio, failure/success statistics of tool calls, and code-pass rates. (Reviewer ybnx W3, Reviewer RCoA W3)
- **Appendix C.2.3 (line 878-899)**: Added efficiency analysis of THOR and the self-correction mechanism. (Reviewer RCoA Q2)

---

### Author Response · Authors · 2025-11-30
**Summary for the Area Chair**

Dear Area Chair,

We sincerely thank all reviewers for their valuable feedback. We are encouraged that our main contributions were positively recognized. Reviewers noted our **novel and effective** RL framework and described the paper as **clearly motivated and technically sound (ybnx, RCoA)**. They highlighted the strength of TIRGen in generating **high-quality, policy-aligned TIR data (ybnx, 5muf)**, and acknowledged that our hierarchical RL design provides **a meaningful and well-justified solution to sparse rewards (5muf, RCoA)**. Reviewers also commended our **strong empirical results and comprehensive ablations (ybnx, 5muf, RCoA)**.

During the rebuttal, we provided point-by-point clarifications and added extensive new experiments in response to all reviewer concerns. Among the three reviewers (ybnx, 5muf, RCoA), reviewer RCoA explicitly stated that **`My concerns have been addressed. And I tend to accept this work.`**

Below is a concise summary of how the rebuttal resolved the issues and questions.

1. Clarifying novelty and contributions (Reviewer ybnx, W4)

   - TIRGen is introduced as **the first unified TIR data-generation pipeline** that robustly supports both reasoning and non-reasoning models.

   - THOR is **the first framework that unifies a fine-grained signal across both hierarchical RL and inference**, enabling stable and scalable tool integration.

2. Strengthened Evidence and Expanded Validation of THOR

   2.1 Evidence of TIRGen Robustness

   - Experiments show that Refiner size has minimal impact on data quality, and manual verification further confirms that TIRGen is robust. (Reviewer 5muf W2, Reviewer RCoA W1)

   2.2 Comparability and Baseline Strengthening

   - We verify that THOR’s training scale is fully comparable to TIR baselines, and we add same-backbone comparisons. (Reviewer ybnx Q4, Q5)

   - We demonstrate strong generalization across tools and downstream tasks. (Reviewer ybnx Q3, Reviewer RCoA W4)

   2.3 Fine-Grained Reward Design

   - We clarify the motivation for using code-pass as the fine-grained reward and provide additional details on the repair strategies. (Reviewer ybnx, W2)

   - We discuss future research directions involving more fine-grained reward shaping for tool calls. (Reviewer ybnx W2, Reviewer 5muf W3)

   2.4 Effectiveness of Self-Correction and Efficiency Analysis

   - New ablations show that step-suffix repair is the most robust repair strategy. (Reviewer ybnx W1&Q1)

   - Failure-mode analysis demonstrates that explicit self-correction significantly reduces execution failures, raising code-pass rates to 98–99%. (Reviewer ybnx W3, Reviewer RCoA W3)

   - Equal-compute comparisons confirm that our self-rewarded inference surpasses strong PRM-based reranking and further improves when combined with PRMs. (Reviewer ybnx Q2)

   - Expanded efficiency analysis shows that self-correction adds negligible overhead and can even reduce total inference cost. (Reviewer RCoA Q2)

3. Explanatory and Terminological Improvements

   - We refine the explanation of advantage computation to improve clarity and reproducibility. (Reviewer RCoA W2, Q1)

   - We revise the names of the key agents in TIRGen in accordance with the reviewers’ suggestions. (Reviewer 5muf W1)


Overall, the rebuttal provides **additional evidence, expanded analysis, and clearer technical explanations aimed at resolving all reviewer concerns**. We hope this summary helps the AC assess the revised submission efficiently.

Best regards,

Authors of submission 5050

---

### Meta-Review · Area_Chair_E39N · 2025-12-28

**Summary:**

This paper proposes THOR, a tool-integrated reasoning framework aimed at improving mathematical reasoning (and related code benchmarks) by combining: (i) TIRGen, a generator–refiner pipeline for producing policy-aligned tool-integrated reasoning (TIR) trajectories; (ii) hierarchical RL that jointly optimizes episode-level final answer correctness and step-level tool/code execution success; and (iii) an inference-time self-correction mechanism that uses tool feedback to backtrack/regenerate when execution fails.

Across both reasoning and non-reasoning model settings, THOR reports strong empirical gains and argues that successful intermediate tool calls are predictive of final correctness, motivating the step-level signal.

The remaining reviewer’s skepticism is largely about repair assumptions, reward analysis, and baseline comparisons; the authors added targeted experiments/analyses that substantially reduce these technical uncertainties.

Overall, THOR’s unified training+inference design around tool feedback (hierarchical RL + self-correction) and the breadth across reasoning/non-reasoning models make this a solid acceptance.

**Reviewer Concerns:**

Concerns:

- Self-correction coupling / repair strategy. Reviewer ybnx questioned the assumption that a failed execution implies both the action and its preceding reasoning are wrong, and asked whether action-only repair might suffice.

- Fine-grained reward choice (code-pass) and analysis of tool-use behavior. Reviewer ybnx and 5muf raised concerns that optimizing only code-pass may be insufficient or may miss important diagnostics (e.g., trigger timing / over-under calling, “logically incorrect but executable” code).

- Comparisons to strong test-time selection / reranking baselines (BoN/self-consistency, PRM/verifiers) and fairness of training scale. Reviewer ybnx explicitly requested these comparisons and checks.

- Reproducibility/clarity and practical overhead. Reviewer RCoA asked for clearer step-level advantage computation and questioned potential inference inefficiency due to evaluation/regeneration.


Addressed:

- Added/expanded repair-strategy ablations; the revised appendix reports that step-suffix repair is the most robust among compared options.

- Added failure-mode/self-correction analysis showing explicit self-correction substantially reduces failed executions and increases code-pass rates (to ~98–99%).

- Added equal-compute comparisons vs self-consistency and PRM/verifier reranking, and showed further gains when combined with reward models in some settings.

- Clarified step-level optimization / advantage computation in the paper.

- Added efficiency analysis indicating self-correction has negligible overhead and overall THOR can be more efficient (shorter generations / improved FPS) in their reported setup.

Outstanding:

- The novelty vs closely related tool-integrated RL lines of work remains a point where the framing could still be debated (ybnx).

- Generalization beyond code as the tool is discussed as future work/limitations; the current scope is largely code-integrated reasoning.

- Reproducibility would be strengthened further by clearer commitments to data/code release (raised by RCoA), though clarity in the algorithmic description improved.

**Reviewer Scores:**

I cannot reliably answer this counterfactual question without putting words in reviewers’ mouths. I will not impute score changes beyond what reviewers explicitly stated in the discussion. I instead provide a faithful synthesis of the discussion outcomes and remaining points of disagreement.

---

### Decision · Program_Chairs · 2026-01-26

Accept (Poster)